# Shared Independent Component Analysis for Multi-Subject Neuroimaging

**Hugo Richard**
Inria
Université Paris-Saclay
Palaiseau, France

**Pierre Ablin**
DMA
CNRS and ENS
Paris, France

**Bertrand Thirion**
Inria
Université Paris-Saclay
Palaiseau, France

**Alexandre Gramfort**
Inria
Université Paris-Saclay
Palaiseau, France

**Aapo Hyvärinen**
Department of Computer Science
University of Helsinki
Helsinki, Finland

## Abstract

We consider shared response modeling, a multi-view learning problem where one wants to identify common components from multiple datasets or views. We introduce Shared Independent Component Analysis (ShICA) that models each view as a linear transform of shared independent components contaminated by additive Gaussian noise. We show that this model is identifiable if the components are either non-Gaussian or have enough diversity in noise variances. We then show that in some cases multi-set canonical correlation analysis can recover the correct unmixing matrices, but that even a small amount of sampling noise makes Multiset CCA fail. To solve this problem, we propose to use joint diagonalization after Multiset CCA, leading to a new approach called ShICA-J. We show via simulations that ShICA-J leads to improved results while being very fast to fit. While ShICA-J is based on second-order statistics, we further propose to leverage non-Gaussianity of the components using a maximum-likelihood method, ShICA-ML, that is both more accurate and more costly. Further, ShICA comes with a principled method for shared components estimation. Finally, we provide empirical evidence on fMRI and MEG datasets that ShICA yields more accurate estimation of the components than alternatives.

## 1 Introduction

In many data science problems, data are available through different views. Generally, the views represent different measurement modalities such as audio and video, or the same text that may be available in different languages. Our main interest here is neuroimaging where recordings are made from multiple subjects. In particular, it is of interest to find common patterns or responses that are shared between subjects when they receive the same stimulation or perform the same cognitive task [16, 51].

A popular line of work to perform such shared response modeling is group Independent Component Analysis (ICA) methods. The fastest methods [14, 58] are among the most popular, yet they are not grounded on principled probabilistic models for the multiview setting. More principled approaches exist [51, 27], but they do not model subject-specific deviations from the shared response. However, such deviations are expected in most neuroimaging settings, as the magnitude of the response may differ from subject to subject [46], as may any noise due to heartbeats, respiratory artefacts or head

35th Conference on Neural Information Processing Systems (NeurIPS 2021).

movements [39]. Furthermore, most GroupICA methods are typically unable to separate components whose density is close to a Gaussian.

Independent vector analysis (IVA) [36, 5] is a powerful framework where components are independent within views but each component of a given view can depend on the corresponding component in other views. However, current implementations such as IVA-L [36], IVA-G [5], IVA-L-SOS [11], IVA-GGD [7] or IVA with Kotz distribution [6] estimate only the view-specific components, and do not model or extract a shared response which is the main focus in this work.

On the other hand, the shared response model [16] is a popular approach to perform shared response modeling, yet it imposes orthogonality constrains that are restrictive and not biologically plausible.

In this work we introduce Shared ICA (ShICA), where each view is modeled as a linear transform of shared independent components contaminated by additive Gaussian noise. ShICA allows the principled extraction of the shared components (or responses) in addition to view-specific components. Since it is based on a statistically sound noise model, it enables optimal inference (minimum mean square error, MMSE) of the shared responses.

Let us note that ShICA is no longer the method of choice when the concept of common response is either not useful or not applicable. Nevertheless, we believe that the ability to extract a common response is an important feature in most contexts because it highlights a stereotypical brain response to a stimulus. Moreover, finding commonality between subjects reduces often unwanted inter-subject variability.

The paper is organized as follows. We first analyse the theoretical properties of the ShICA model, before providing inference algorithms. We exhibit necessary and sufficient conditions for the ShICA model to be identifiable (previous work only shows local identifiability [7]), in the presence of Gaussian or non-Gaussian components. We then use Multiset CCA to fit the model when all the components are assumed to be Gaussian. We exhibit necessary and sufficient conditions for Multiset CCA to be able to recover the unmixing matrices (previous work only gives sufficient conditions [38]). In addition, we provide instances of the problem where Multiset CCA cannot recover the mixing matrices while the model is identifiable. We next point out a practical problem : even a small sampling noise can lead to large error in the estimation of unmixing matrices when Multiset CCA is used. To address this issue and recover the correct unmixing matrices, we propose to apply joint diagonalization to the result of Multiset CCA yielding a new method called ShICA-J. We further introduce ShICA-ML, a maximum likelihood estimator of ShICA that models non-Gaussian components using a Gaussian mixture model. While ShICA-ML yields more accurate components, ShICA-J is significantly faster and offers a great initialization to ShICA-ML. Experiments on fMRI and MEG data demonstrate that the method outperforms existing GroupICA and IVA methods.

## 2    Shared ICA (ShICA): an identifiable multi-view model

**Notation**    We write vectors in bold letter $\mathbf{v}$ and scalars in lower case $a$. Upper case letters $M$ are used to denote matrices. We denote $|M|$ the absolute value of the determinant of $M$. $\mathbf{x} \sim \mathcal{N}(\boldsymbol{\mu}, \Sigma)$ means that $\mathbf{x} \in \mathbb{R}^k$ follows a multivariate normal distribution of mean $\boldsymbol{\mu} \in \mathbb{R}^k$ and covariance $\Sigma \in \mathbb{R}^{k \times k}$. The $j, j$ entry of a diagonal matrix $\Sigma_i$ is denoted $\Sigma_{ij}$, the $j$ entry of $\mathbf{y}_i$ is denoted $y_{ij}$. Lastly, $\delta$ is the Kronecker delta.

**Model Definition**    In the following, $\mathbf{x}_1, \ldots, \mathbf{x}_m \in \mathbb{R}^p$ denote the $m$ observed random vectors obtained from the $m$ different views. We posit the following generative model, called Shared ICA (ShICA): for $i = 1 \ldots m$

$$\mathbf{x}_i = A_i(\mathbf{s} + \mathbf{n}_i) \tag{1}$$

where $\mathbf{s} \in \mathbb{R}^p$ contains the latent variables called *shared components*, $A_1, \ldots, A_m \in \mathbb{R}^{p \times p}$ are the invertible mixing matrices, and $\mathbf{n}_i \in \mathbb{R}^p$ are *individual noises*. The individual noises model both the deviations of a view from the mean —i.e. individual differences— and measurement noise. Importantly, we explicitly model both the shared components and the individual differences in a probabilistic framework to enable an optimal inference of the parameters and the responses.

We assume that the shared components are statistically independent, and that the individual noises are Gaussian and independent from the shared components: $p(\mathbf{s}) = \prod_{j=1}^p p(s_j)$ and $\mathbf{n}_i \sim \mathcal{N}(0, \Sigma_i)$, where the matrices $\Sigma_i$ are assumed diagonal and positive. Without loss of generality, components

are assumed to have unit variance $\mathbb{E}[\mathbf{s}\mathbf{s}^\top] = I_p$. We further assume that there are at least 3 views: $m \geq 3$.

In contrast to almost all existing works, we assume that some components (possibly all of them) may be Gaussian, and denote $\mathcal{G}$ the set of Gaussian components: $\mathbf{s}_j \sim \mathcal{N}(0,1)$ for $j \in \mathcal{G}$. The other components are non-Gaussian: for $j \notin \mathcal{G}$, $\mathbf{s}_j$ is non-Gaussian.

**Identifiability**  The parameters of the model are $\Theta = (A_1, \dots, A_m, \Sigma_1, \dots, \Sigma_m)$. We are interested in the identifiability of this model: given observations $\mathbf{x}_1, \dots, \mathbf{x}_m$ generated with parameters $\Theta$, are there some other $\Theta'$ that may generate the same observations? Let us consider the following assumption that requires that the individual noises for Gaussian components are sufficiently diverse:

**Assumption 1** (Noise diversity in Gaussian components). *For all $j, j' \in \mathcal{G}, j \neq j'$, the sequences $(\Sigma_{ij})_{i=1\dots m}$ and $(\Sigma_{ij'})_{i=1\dots m}$ are different where $\Sigma_{ij}$ is the $j, j$ entry of $\Sigma_i$*

It is readily seen that there is one trivial set of indeterminacies in the problem: if $P \in \mathbb{R}^{p \times p}$ is a sign and permutation matrix (i.e. a matrix which has one $\pm 1$ coefficient on each row and column, and 0's elsewhere) the parameters $(A_1 P, \dots, A_m P, P^\top \Sigma_1 P, \dots, P^\top \Sigma_m P)$ also generate $\mathbf{x}_1, \dots, \mathbf{x}_m$. The following theorem shows that under the above assumption, these are the only indeterminacies of the problem.

**Theorem 1** (Identifiability). *We make Assumption 1. We let $\Theta' = (A'_1, \dots, A'_m, \Sigma'_1, \dots, \Sigma'_m)$ another set of parameters, and assume that they also generate $\mathbf{x}_1, \dots, \mathbf{x}_m$. Then, there exists a sign and permutation matrix $P$ such that for all $i$, $A'_i = A_i P$, and $\Sigma'_i = P^\top \Sigma_i P$.*

The proof is in Appendix A.1. Identifiability in the Gaussian case is a consequence of the identifiability results in [59] and in the general case, local identifiability results can be derived from the work of [7]. However local identifiability only shows that for a given set of parameters there exists a neighborhood in which no other set of parameters can generate the same observations [52]. In contrast, the proof of Theorem 1 shows global identifiability.

Theorem 1 shows that the task of recovering the parameters from the observations is a well-posed problem, under the sufficient condition of Assumption 1. We also note that Assumption 1 is necessary for identifiability. For instance, if $j$ and $j'$ are two Gaussian components such that $\Sigma_{ij} = \Sigma_{ij'}$ for all $i$, then a global rotation of the components $j, j'$ yields the same covariance matrices. The current work assumes $m \geq 3$, in appendix B we give an identifiability result for $m = 2$, under stronger conditions.

## 3 Estimation of components with noise diversity via joint-diagonalization

We now consider the computational problem of efficient parameter inference. This section considers components with noise diversity, while the next section deals with non-Gaussian components.

### 3.1 Parameter estimation with Multiset CCA

If we assume that the components are all Gaussian, the covariance of the observations given by $C_{ij} = \mathbb{E}[\mathbf{x}_i \mathbf{x}_j^\top] = A_i(I_p + \delta_{ij}\Sigma_i)A_j^\top$ are sufficient statistics and methods using only second order information, like Multiset CCA, are candidates to estimate the parameters of the model. Consider the matrix $\mathcal{C} \in \mathbb{R}^{pm \times pm}$ containing $m \times m$ blocks of size $p \times p$ such that the block $i, j$ is given by $C_{ij}$. Consider the matrix $\mathcal{D}$ identical to $\mathcal{C}$ excepts that the non-diagonal blocks are filled with zeros:

$$\mathcal{C} = \begin{bmatrix} C_{11} & \dots & C_{1m} \\ \vdots & \ddots & \vdots \\ C_{m1} & \dots & C_{mm} \end{bmatrix}, \ \mathcal{D} = \begin{bmatrix} C_{11} & \dots & 0 \\ \vdots & \ddots & \vdots \\ 0 & \dots & C_{mm} \end{bmatrix}. \tag{2}$$

Generalized CCA consists of the following generalized eigenvalue problem:

$$\mathcal{C}\mathbf{u} = \lambda \mathcal{D}\mathbf{u}, \ \lambda > 0, \ \mathbf{u} \in \mathbb{R}^{pm} \ . \tag{3}$$

Consider the matrix $U = [\mathbf{u}^1, \dots, \mathbf{u}^p] \in \mathbb{R}^{mp \times p}$ formed by concatenating the $p$ leading eigenvectors of the previous problem ranked in decreasing eigenvalue order. Then, consider $U$ to be formed of $m$

blocks of size $p \times p$ stacked vertically and define $(W^i)^\top$ to be the $i$-th block. These $m$ matrices are the output of Multiset CCA. We also denote $\lambda_1 \geq \cdots \geq \lambda_p$ the $p$ leading eigenvalues of the problem.

An application of the results of [38] shows that Multiset CCA recovers the mixing matrices of ShICA under some assumptions.

**Proposition 1** (Sufficient condition for solving ShICA via Multiset CCA [38]). *Let $r_{ijk} = (1 + \Sigma_{ik})^{-\frac{1}{2}}(1 + \Sigma_{jk})^{-\frac{1}{2}}$. Assume that $(r_{ijk})_k$ is non-increasing. Assume that the maximum eigenvalue $\nu_k$ of matrix $R^{(k)}$ of general element $(r_{ijk})_{ij}$ is such that $\nu_k = \lambda_k$. Assume that $\lambda_1 \ldots \lambda_p$ are distinct. Then, there exists scale matrices $\Gamma_i$ such that $W_i = \Gamma_i A_i^{-1}$ for all $i$.*

This proposition gives a sufficient condition for solving ShICA with Multiset CCA. It needs a particular structure for the noise covariances as well as specific ordering for the eigenvalues. The next theorem shows that we only need $\lambda_1 \ldots \lambda_p$ to be distinct for Multiset CCA to solve ShICA:

**Assumption 2** (Unique eigenvalues). $\lambda_1 \ldots \lambda_p$ *are distinct.*

**Theorem 2.** *We only make Assumption 2. Then, there exists a permutation matrix $P$ and scale matrices $\Gamma_i$ such that $W_i = P\Gamma_i A_i^{-1}$ for all $i$.*

The proof is in Appendix A.2. This theorem means that solving the generalized eigenvalue problem (3) allows to recover the mixing matrices up to a scaling and permutation: this form of generalized CCA recovers the parameters of the statistical model. Note that Assumption 2 is also a necessary condition. Indeed, if two eigenvalues are identical, the eigenvalue problem is not uniquely determined.

We have two different Assumptions, 1 and 2, the first of which guarantees theoretical identifiability as per Theorem 1 and the second guarantees consistent estimation by Multiset CCA as per Theorem 2. Next we will discuss their connections, and show some limitations of the Multiset CCA approach. To begin with, we have the following result about the eigenvalues of the problem (3) and the $\Sigma_{ij}$.

**Proposition 2.** *For $j \leq p$, let $\lambda_j$ the largest solution of $\sum_{i=1}^{m} \frac{1}{\lambda_j(1+\Sigma_{ij})-\Sigma_{ij}} = 1$. Then, $\lambda_1, \ldots, \lambda_p$ are the $p$ largest eigenvalues of problem* (3).

It is easy to see that we then have $\lambda_1, \ldots, \lambda_p$ greater than 1, while the remaining eigenvalues are lower than 1. From this proposition, two things appear clearly. First, Assumption 2 implies Assumption 1. Indeed, if the $\lambda_j$'s are distinct, then the sequences $(\Sigma_{ij})_i$ must also be different from the previous proposition. This is expected as from Theorem 2, Assumption 2 implies identifiability, which in turn implies Assumption 1.

Prop. 2 also allows us to derive cases where Assumption 1 holds but not Assumption 2. The following Proposition gives a simple case where the model is identifiable but it cannot be solved using Multiset CCA:

**Proposition 3.** *Assume that for two integers $j, j'$, the sequence $(\Sigma_{ij})_i$ is a permutation of $(\Sigma_{ij'})_i$, i.e. that there exists a permutation of $\{1, \ldots, p\}$, $\pi$, such that for all $i$, $\Sigma_{ij} = \Sigma_{\pi(i)j'}$. Then, $\lambda_j = \lambda_{j'}$.*

In this setting, Assumption 1 holds so ShICA is identifiable, while Assumption 2 does not hold, so Multiset CCA cannot recover the unmixing matrices.

## 3.2 Sampling noise and improved estimation with joint diagonalization

The consistency theory for Multiset CCA developed above is conducted under the assumption that the covariances $C_{ij}$ are the true covariances of the model, and not approximations obtained from observed samples. In practice, however, a serious limitation of Multiset CCA is that even a slight error of estimation on the covariances, due to "sampling noise", can yield a large error in the estimation of the unmixing matrices, as will be shown next.

We begin with an empirical illustration. We take $m = 3$, $p = 2$, and $\Sigma_i$ such that $\lambda_1 = 2 + \varepsilon$ and $\lambda_2 = 2$ for $\varepsilon > 0$. In this way, we can control the *eigen-gap* of the problem, $\varepsilon$. We take $W_i$ the outputs of Multiset CCA applied to the true covariances $C_{ij}$. Then, we generate a perturbation $\Delta = \delta \cdot S$, where $S$ is a random positive symmetric $pm \times pm$ matrix of norm 1, and $\delta > 0$ controls the scale of the perturbation. We take $\Delta_{ij}$ the $p \times p$ block of $\Delta$ in position $(i, j)$, and $\tilde{W}_i$ the output of Multiset CCA applied to the covariances $C_{ij} + \Delta_{ij}$. We finally compute the sum of the Amari distance between the $W_i$ and $\tilde{W}_i$: the Amari distance measures how close the two matrices are, up to scale and permutation [4].

Fig 1 displays the median Amari distance over 100 random repetitions, as the perturbation scale $\delta$ increases. The different curves correspond to different values of the eigen-gap $\varepsilon$. We see clearly that the robustness of Multiset CCA critically depends on the eigen-gap, and when it is small, even a small perturbation of the input (due, for instance, to sampling noise) leads to large estimation errors.

This problem is very general and well studied [53]: the mapping from matrices to (generalized) eigenvectors is highly non-smooth. However, the gist of our method is that the *span* of the leading $p$ eigenvectors is smooth, as long as there is a large enough gap between $\lambda_p$ and $\lambda_{p+1}$. For our specific problem we have the following bounds, derived from Prop. 2.

**Proposition 4.** *We let $\sigma_{\max} = \max_{ij} \Sigma_{ij}$ and $\sigma_{\min} = \min_{ij} \Sigma_{ij}$. Then, $\lambda_p \geq 1 + \frac{m-1}{1+\sigma_{\max}}$, while $\lambda_{p+1} \leq 1 - \frac{1}{1+\sigma_{min}}$.*

As a consequence, we have $\lambda_p - \lambda_{p+1} \geq \frac{m-1}{1+\sigma_{\max}} + \frac{1}{1+\sigma_{\min}} \geq \frac{m}{1+\sigma_{\max}}$: the gap between these eigenvalues increases with $m$, and decreases with the noise power.

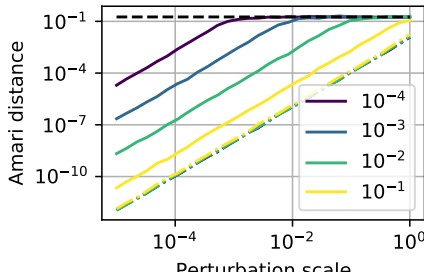

Figure 1: Amari distance between true mixing matrices and estimates of Multi-set CCA when covariances are perturbed. Different solid curves correspond to different eigen-gaps. The black dotted line shows the chance level. When the gap is small, a small perturbation can lead to complete mixing. Joint-diagonalization (colored dotted lines) fixes the problem.

In this setting, when the magnitude of the perturbation $\Delta$ is smaller than $\lambda_p - \lambda_{p+1}$, [53] indicates that $\mathrm{Span}([W_1, \ldots, W_m]^\top) \simeq \mathrm{Span}([\tilde{W}_1, \ldots, \tilde{W}_m]^\top)$, where $[W_1, \ldots, W_m]^\top \in \mathbb{R}^{pm \times p}$ is the vertical concatenation of the $W_i$'s. In turn, this shows that there exists a matrix $Q \in \mathbb{R}^{p \times p}$ such that

$$W_i \simeq Q\tilde{W}_i \text{ for all } i. \tag{4}$$

We propose to use joint-diagonalization to recover the matrix $Q$. Given the $\tilde{W}_i$'s, we consider the set of symmetric matrices $\tilde{K}_i = \tilde{W}_i \tilde{C}_{ii} \tilde{W}_i^\top$, where $\tilde{C}_{ii}$ is the contaminated covariance of $\mathbf{x}_i$. Following Eq. (4), we have $Q\tilde{K}_i Q^\top = W_i \tilde{C}_{ii} W_i^\top$, and using Theorem 2, we have $Q\tilde{K}_i Q^\top = P\Gamma_i A_i^{-1} \tilde{C}_{ii} A_i^{-\top} \Gamma_i P^\top$. Since $\tilde{C}_{ii}$ is close to $C_{ii} = A_i(I_p + \Sigma_i)A_i^\top$, the matrix $P\Gamma_i A_i^{-1} \tilde{C}_{ii} A_i^{-\top} \Gamma_i P^\top$ is almost diagonal. In other words, the matrix $Q$ is an approximate diagonalizer of the $\tilde{K}_i$'s, and we approximate $Q$ by joint-diagonalization of the $\tilde{K}_i$'s. In Fig 1, we see that this procedure mitigates the problems of multiset-CCA, and gets uniformly better performance regardless of the eigen-gap. In practice, we use a fast joint-diagonalization algorithm [1] to minimize a joint-diagonalization criterion for positive symmetric matrices [48]. The estimated unmixing matrices $U_i = Q\tilde{W}_i$ correspond to the true unmixing matrices only up to some scaling which may be different from subject to subject: the information that the components are of unit variance is lost. As a consequence, naive averaging of the recovered components may lead to inconsistant estimation. We now describe a procedure to recover the correct scale of the individual components across subjects.

**Algorithm 1** ShICA-J

---

**Input :** Covariances $\tilde{C}_{ij} = \mathbb{E}[\mathbf{x}_i \mathbf{x}_j^\top]$
$(\tilde{W}_i)_i \leftarrow \mathrm{MultisetCCA}((\tilde{C}_{ij})_{ij})$
$Q \leftarrow \mathrm{JointDiag}((\tilde{W}_i \tilde{C}_{ii} \tilde{W}_i^\top)_i)$
$\Gamma_{ij} \leftarrow Q\tilde{W}_i \tilde{C}_{ij} W_j^\top Q^\top$
$(\Phi_i)_i \leftarrow \mathrm{Scaling}((\Gamma_{ij})_{ij})$
**Return :** Unmixing matrices $(\Phi_i Q\tilde{W}_i)_i$.

---

**Scale estimation** We form the matrices $\Gamma_{ij} = U_i \tilde{C}_{ij} U_j^\top$. In order to estimate the scalings, we solve $\min_{(\Phi_i)} \sum_{i \neq j} \|\Phi_i \mathrm{diag}(\Gamma_{ij})\Phi_j - I_p\|_F^2$ where the $\Phi_i$ are diagonal matrices. This function is readily minimized with respect to one of the $\Phi_i$ by the formula $\Phi_i = \frac{\sum_{j \neq i} \Phi_j \mathrm{diag}(Y_{ij})}{\sum_{j \neq i} \Phi_j^2 \mathrm{diag}(Y_{ij})^2}$ (derivations in Appendix 20). We then iterate the previous formula over $i$ until convergence. The final estimates of the unmixing matrices are given by $(\Phi_i U_i)_{i=1}^m$. The full procedure, called ShICA-J, is summarized in Algorithm 1.

### 3.3 Estimation of noise covariances

In practice, it is important to estimate noise covariances $\Sigma_i$ in order to take advantage of the fact that

some views are noisier than others. As it is well known in classical factor analysis, modelling noise variances allows the model to virtually discard variables, or subjects, that are particularly noisy.

Using the ShICA model with Gaussian components, we derive an estimate for the noise covariances directly from maximum likelihood. We use an expectation-maximization (EM) algorithm, which is especially fast because noise updates are in closed-form. Following derivations given in appendix D.1, the sufficient statistics in the E-step are given by

$$\mathbb{E}[\mathbf{s}|\mathbf{x}] = \left(\sum_{i=1}^m \Sigma_i^{-1} + I\right)^{-1} \sum_{i=1}^m \left(\Sigma_i^{-1}\mathbf{y}_i\right) \qquad \mathbb{V}[\mathbf{s}|\mathbf{x}] = (\sum_{i=1}^m \Sigma_i^{-1} + I)^{-1} \qquad (5)$$

Incorporating the M-step we get the following updates that only depend on the covariance matrices:
$$\Sigma_i \leftarrow \mathrm{diag}(\hat{C}_{ii} - 2\mathbb{V}[\mathbf{s}|\mathbf{x}]\sum_{j=1}^m \Sigma_j^{-1}\hat{C}_{ji} + \mathbb{V}[\mathbf{s}|\mathbf{x}]\sum_{j=1}^m\sum_{l=1}^m \left(\Sigma_j^{-1}\hat{C}_{jl}\Sigma_l^{-1}\right)\mathbb{V}[\mathbf{s}|\mathbf{x}] + \mathbb{V}[\mathbf{s}|\mathbf{x}])$$

## 4 ShICA-ML: Maximum likelihood for non-Gaussian components

ShICA-J only uses second order statistics. However, the ShICA model (1) allows for non-Gaussian components. We now propose an algorithm for fitting the ShICA model that combines covariance information with non-Gaussianity in the estimation to optimally separate both Gaussian and non-Gaussian components. We estimate the parameters by maximum likelihood. Since most non-Gaussian components in real data are super-Gaussian [21, 13], we assume that the non-Gaussian components $\mathbf{s}$ have the super-Gaussian density
$p(s_j) = \frac{1}{2}\left(\mathcal{N}(s_j; 0, \frac{1}{2}) + \mathcal{N}(s_j; 0, \frac{3}{2})\right)$ .

We propose to maximize the log-likelihood using a generalized EM [41, 22]. Derivations are available in Appendix E. Like in the previous section, the E-step is in closed-form yielding the following sufficient statistics:

$$\mathbb{E}[s_j|\mathbf{x}] = \frac{\sum_{\alpha\in\{\frac{1}{2},\frac{3}{2}\}} \theta_\alpha \frac{\alpha\bar{y}_j}{\alpha+\bar{\Sigma}_j}}{\sum_{\alpha\in\{0.5,1.5\}} \theta_\alpha} \quad \text{and} \quad \mathbb{V}[s_j|\mathbf{x}] = \frac{\sum_{\alpha\in\{\frac{1}{2},\frac{3}{2}\}} \theta_\alpha \frac{\bar{\Sigma}_j\alpha}{\alpha+\bar{\Sigma}_j}}{\sum_{\alpha\in\{0.5,1.5\}} \theta_\alpha} \qquad (6)$$

where $\theta_\alpha = \mathcal{N}(\bar{y}_j; 0, \bar{\Sigma}_j + \alpha)$, $\bar{y}_j = \frac{\sum_i \Sigma_{ij}^{-1}y_{ij}}{\sum_i \Sigma_{ij}^{-1}}$ and $\bar{\Sigma}_j = (\sum_i \Sigma_{ij}^{-1})^{-1}$ with $\mathbf{y}_i = W_i\mathbf{x}_i$. Noise updates are in closed-form and given by: $\Sigma_i \leftarrow \mathrm{diag}((\mathbf{y}_i - \mathbb{E}[\mathbf{s}|\mathbf{x}])(\mathbf{y}_i - \mathbb{E}[\mathbf{s}|\mathbf{x}])^\top + \mathbb{V}[\mathbf{s}|\mathbf{x}])$. However, no closed-form is available for the updates of unmixing matrices. We therefore perform quasi-Newton updates given by $W_i \leftarrow (I - \rho(\widehat{\mathcal{H}^{W_i}})^{-1}\mathcal{G}^{W_i})W_i$ where $\rho \in \mathbb{R}$ is chosen by backtracking line-search, $\widehat{\mathcal{H}^{W_i}}_{a,b,c,d} = \delta_{ad}\delta_{bc} + \delta_{ac}\delta_{bd}\frac{(y_{ib})^2}{\Sigma_{ia}}$ is an approximation of the Hessian of the negative complete likelihood and $\mathcal{G}^{W_i} = -I + (\Sigma_i)^{-1}(\mathbf{y}_i - \mathbb{E}[\mathbf{s}|\mathbf{x}])(\mathbf{y}_i)^\top$ is the gradient.

We alternate between computing the statistics $\mathbb{E}[\mathbf{s}|\mathbf{x}]$, $\mathbb{V}[\mathbf{s}|\mathbf{x}]$ (E-step) and updates of parameters $\Sigma_i$ and $W_i$ for $i = 1 \ldots m$ (M-step). Let us highlight that our EM algorithm and in particular the E-step resembles the one used in [40]. However because they assume noise on the sensors and not on the components, their formula for $\mathbb{E}[\mathbf{s}|\mathbf{x}]$ involves a sum with $2^p$ terms whereas we have only 2 terms. The resulting method is called ShICA-ML.

**Minimum mean square error estimates in ShICA** In ShICA-J as well as in ShICA-ML, we have a closed-form for the expected components given the data $\mathbb{E}[\mathbf{s}|\mathbf{x}]$, shown in equation (5) and (6) respectively. This provides minimum mean square error estimates of the shared components, and is an important benefit of explicitly modelling shared components in a probabilistic framework.

## 5 Related Work

ShICA combines theory and methods coming from different branches of "component analysis". It can be viewed as a GroupICA method, as an extension of Multiset CCA, as an Independent Vector Analysis method or, crucially, as an extension of the shared response model. In the setting studied here, ShICA improves upon all existing methods.

**GroupICA** GroupICA methods extract independent components from multiple datasets. In its original form[14], views are concatenated and then a PCA is applied yielding reduced data on which ICA is applied. One can also reduce the data using Multiset CCA instead of PCA, giving a method called *CanICA* [58]. Other works [24, 32] apply ICA separately on the datasets and attempt to match the decompositions afterwards. Although these works provide very fast methods, they do not rely on a well defined model like ShICA. Other GroupICA methods impose some structure on the mixing matrices such as the tensorial method of [10] or the group tensor model in [27] (which assumes identical mixing matrices up to a scaling) or [54] (which assumes identical mixing matrices but different components). In ShICA the mixing matrices are only constrained to be invertible. Lastly, maximum-likelihood based methods exist such as *MultiViewICA* [51] (MVICA) or the full model of [27]. These methods are weaker than ShICA as they use the same noise covariance across views and lack a principled method for shared response inference.

**Multiset CCA** In its basic formulation, CCA identifies a shared space between two datasets. The extension to more than two datasets is ambiguous, and many different generalized CCA methods have been proposed. [33] introduces 6 objective functions that reduce to CCA when $m = 2$ and [42] considered 4 different possible constrains leading to 24 different formulations of Multiset CCA. The formulation used in ShICA-J is refered to in [42] as SUMCORR with constraint 4 which is one of the fastest as it reduces to solving a generalized eigenvalue problem. The fact that CCA solves a well defined probabilistic model has first been studied in [9] where it is shown that CCA is identical to multiple battery factor analysis [12] (restricted to 2 views). This latter formulation differs from our model in that the noise is added on the sensors and not on the components which makes the model unidentifiable. Identifiable variants and generalizations can be obtained by imposing sparsity on the mixing matrices such as in [8, 34, 62] or non-negativity [20]. The work in [38] exhibits a set of sufficient (but not necessary) conditions under which a well defined model can be learnt by the formulation of Multiset CCA used in ShICA-J. The set of conditions we exhibit in this work are necessary and sufficient. We further emphasize that basic Multiset CCA provides a poor estimator as explained in Section 3.2.

**Independent vector analysis** Independent vector analysis [36] (IVA) models the data as a linear mixture of independent components $\mathbf{x}_i = A_i \mathbf{s}_i$ where each component $s_{ij}$ of a given view $i$ can depend on the corresponding component in other views $((s_{ij})_{i=1}^m$ are not independent). Practical implementations of this very general idea assume a distribution for $p((s_{ij})_{i=1}^m)$. In IVA-L [36], $p((s_{ij})_{i=1}^m) \propto \exp(-\sqrt{\sum_i (s_{ij})^2})$ (so the variance of each component in each view is assumed to be the same), in IVA-G [5] or in [60], $p((s_{ij})_{i=1}^m) \sim \mathcal{N}(0, R_{ss})$ and [23] proposed a normal inverse-Gamma density. Let us also mention IVA-L-SOS [11], IVA-GGD [7] and IVA with Kotz distribution [6] that assume a non-Gaussian density general enough so that they can use both second and higher order statistics to extract view-specific components. The model of ShICA can be seen as an instance of IVA which specifically enables extraction of shared components from the subject specific components, unlike previous versions of IVA. In fact, ShICA comes with minimum mean square error estimates for the shared components that is often the quantity of interest. The IVA theory provides global identifiability conditions in the Gaussian case (IVA-G) [59] and local identifiability conditions in the general case [7] from which local identifiability conditions of ShICA could be derived. However, in this work, we provide global identifiability conditions for ShICA. Lastly, IVA can be performed using joint diagonalization of cross covariances [37, 19] although multiple matrices have to be learnt and cross-covariances are not necessarily symmetric positive definite which makes the algorithm slower and less principled.

**Shared response model** ShICA extracts shared components from multiple datasets, which is also the goal of the shared response model (SRM) [16]. The robust SRM [57] also allows to capture subject specific noise. However these models impose orthogonality constraints on the mixing matrices while ShICA does not. Deep variants of SRM exist such as [17] but while they release the orthogonality constrain, they are not very easy to train or interpret and have many hyper-parameters to tune. ShICA leverages ICA theory to provide a much more powerful model of shared responses.

**Limitations** The main limitation of this work is that the model cannot reduce the dimension inside each view : there are as many estimated sources as sensors. This might be problematic when the number of sensors is very high. In line with other methods, view-specific dimension reduction has to be done by some external method, typically view-specific PCA. Using specialized methods

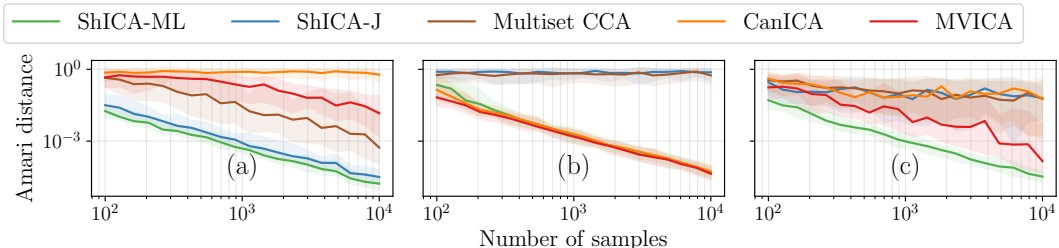

Figure 2: **Separation performance**: Algorithms are fit on data following model 1 **(a)** Gaussian components with noise diversity **(b)** Non-Gaussian components without noise diversity **(c)** Half of the components are Gaussian with noise diversity, the other half is non-Gaussian without noise diversity.

for the estimation of covariances should also be of interest for ShICA-J, where it only relies on sample covariances. Finally, ShICA-ML uses a simple model of a super-Gaussian distribution, while modelling the non-gaussianities in more detail in ShICA-ML should improve the performance.

## 6 Experiments

Experiments used Nilearn [3] and MNE [26] for fMRI and MEG data processing respectively, as well as the scientific Python ecosystem: Matplotlib [31], Scikit-learn [45], Numpy [29] and Scipy [61]. We use the Picard algorithm for non-Gaussian ICA [2], and mvlearn for multi-view ICA [47]. The above libraries use open-source licenses. fMRI experiments used the following datasets: sherlock [15], forrest [28] , raiders [49] and gallant [49]. The data we use do not contain offensive content or identifiable information and consent was obtained before data collection. Computations were run on a large server using up to 100 GB of RAM and 20 CPUs in parallel.

**Separation performance** In the following synthetic experiments, data are generated according to model (1) with $p = 4$ components and $m = 5$ views and mixing matrices are generated by sampling coefficients from a standardized Gaussian. Gaussian components are generated from a standardized Gaussian and their noise has standard deviation $\Sigma_i^{\frac{1}{2}}$ (obtained by sampling from a uniform density between 0 and 1) while non-Gaussian components are generated from a Laplace distribution and their noise standard deviations are equal. We study 3 cases where either all components are Gaussian, all components are non-Gaussian or half of the components are Gaussian and half are non-Gaussian. We vary the number of samples $n$ between $10^2$ and $10^5$ and display in Fig 2 the mean Amari distance across subjects between the true unmixing matrices and estimates of algorithms as a function of $n$. The experiment is repeated 100 times using different seeds. We report the median result and error bars represent the first and last deciles.

When all components are Gaussian (Fig. 2 (a)), CanICA cannot separate the components at all. In contrast ShICA-J, ShICA-ML, Multiset CCA and MVICA are able to separate them, but Multiset CCA needs many more samples than ShICA-J or ShICA-ML to reach a low amari distance, which shows that correcting for the rotation due to sampling noise improves the results. Looking at error bars, we also see that the performance of Multiset CCA varies quite a lot with the random seeds: this shows that depending on the sampling noise, the rotation can be very different from identity. MVICA needs even more sample than Multiset CCA to reach a low amari distance but still outperforms CanICA.

When none of the components are Gaussian (Fig. 2 (b)), only CanICA, ShICA-ML and MVICA are able to separate the components, as other methods do not make use of non-Gaussianity. Finally, in the hybrid case (Fig. 2 (c)), ShICA-ML is able to separate the components as it can make use of both non-Gaussianity and noise diversity. MVICA is a lot less reliable than ShICA-ML, it is uniformly worse and error bars are very large showing that for some seeds it gives poor results. CanICA, ShICA-J and MultisetCCA cannot separate the components at all. Additional experiments illustrating the separation powers of algorithms are available in Appendix H.1.

As we can see, MVICA can separate Gaussian components to some extent and therefore does not completely fail when Gaussian and non-Gaussian components are present. However MVICA is a lot

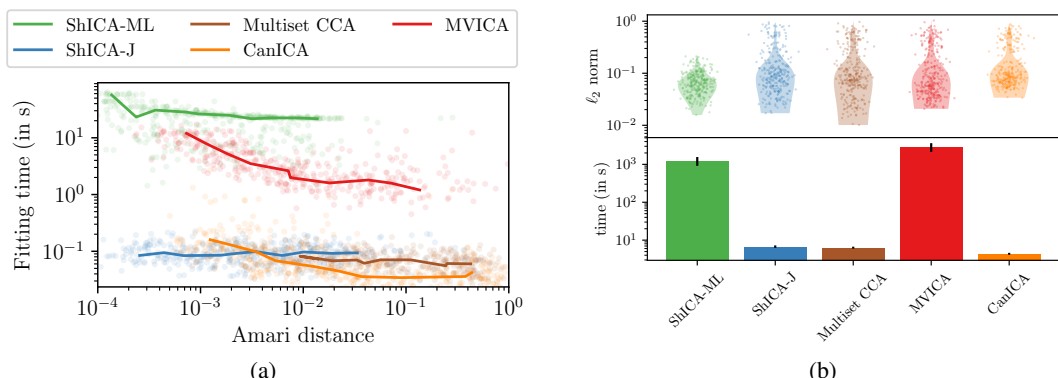

(a)                                        (b)

Figure 3: **Left: Computation time.** Algorithms are fit on data generated from model (1) with a super-Gaussian density. For different values of the number of samples, we plot the Amari distance and the fitting time. Thick lines link median values across seeds. **Right: Robustness w.r.t intra-subject variability in MEG.** (**top**) $\ell_2$ distance between shared components corresponding to the same stimuli in different trials. (**bottom**) Fitting time.

less reliable than ShICA-ML: MVICA is uniformly worse than ShICA-ML and the error bars are very large showing that for some seeds it gives poor results.

**Computation time** We generate components using a slightly super Gaussian density: $s_j = d(x)$ with $d(x) = x|x|^{0.2}$ and $x \sim \mathcal{N}(0,1)$. We vary the number of samples $n$ between $10^2$ and $10^4$. We compute the mean Amari distance across subjects and record the computation time. The experiment is repeated 40 times. We plot the Amari distance as a function of the computation time in Fig 3a. Each point corresponds to the Amari distance/computation time for a given number of samples and a given seed. We then consider for a given number of samples, the median Amari distance and computation time across seeds and plot them in the form of a thick line. From Fig 3a, we see that ShICA-J is the method of choice when speed is a concern while ShICA-ML yields the best performance in terms of Amari distance at the cost of an increased computation time. The thick lines for ShICA-J and Multiset CCA are quasi-flat, indicating that the number of samples does not have a strong impact on the fitting time as these methods only work with covariances. On the other hand CanICA or MVICA computation time is more sensitive to the number of samples.

**Robustness w.r.t intra-subject variability in MEG** In the following experiments we consider the Cam-CAN dataset [56]. We use the magnetometer data from the MEG of $m = 100$ subjects chosen randomly among 496. In appendix F we give more information about Cam-CAN dataset. Each subject is repeatedly presented three audio-visual stimuli. For each stimulus, we divide the trials into two sets and within each set, the MEG signal is averaged across trials to isolate the evoked response. This procedure yields 6 chunks of individual data (2 per stimulus). We study the similarity between shared components corresponding to repetitions of the same stimulus. This gives a measure of robustness of each ICA algorithm with respect to intra-subject variability. Data are first reduced using a subject-specific PCA with $p = 10$ components. The initial dimensionality of the data before PCA is 102 as we only use the 102 magnetometers. Algorithms are run 10 times with different seeds on the 6 chunks of data, and shared components are extracted. When two chunks of data correspond to repetitions of the same stimulus they should yield similar components. For each component and for each stimulus, we therefore measure the $\ell_2$ distance between the two repetitions of the stimulus. This yields 300 distances per algorithm that are plotted on Fig 3b.

The components recovered by ShICA-ML have a much lower variability than other approaches. The performance of ShICA-J is competitive with MVICA while being much faster to fit. Multiset CCA yields satisfying results compared with ShICA-J. However we see that the number of components that do not match at all across trials is greater in Multiset CCA. Additional experiments on MEG data are available in Appendix H.3.

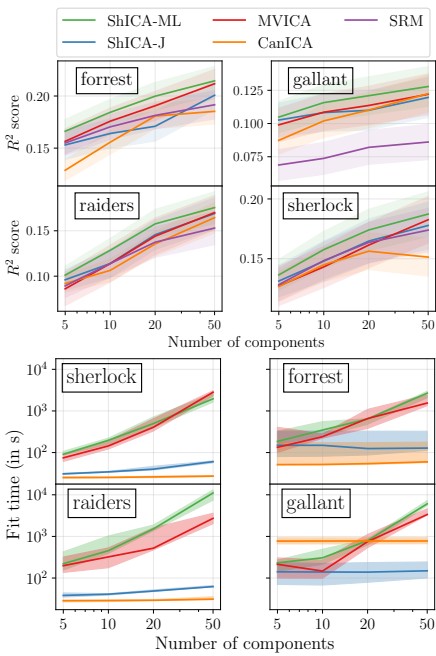

Figure 4: **Reconstructing the BOLD signal of missing subjects**. (**top**) Mean $R^2$ score between reconstructed data and true data. (**bottom**) Fitting time.

**Reconstructing the BOLD signal of missing subjects**
We reproduce the experimental pipeline of [51] to benchmark GroupICA methods using their ability to reconstruct fMRI data of a left-out subject. The preprocessing involves a dimension reduction step performed using the shared response model [16]. Detailed preprocessing pipeline is described in Appendix F. We call an *unmixing operator* the product of the dimension reduction operator and an unmixing matrix and a *mixing operator* its pseudoinverse. There is one unmixing operator and one mixing operator per view. The unmixing operators are learned using all subjects and $80\%$ of the runs. Then they are applied on the remaining $20\%$ of the runs using $80\%$ of the subjects yielding unmixed data from which shared components are extracted. The unmixed data are combined by averaging (for SRM and other baselines) or using the MMSE estimate for ShICA-J and ShICA-ML. We then apply the mixing operator of the remaining $20\%$ subjects on the shared components to reconstruct their data. Reconstruction accuracy is measured via the coefficient of determination, *a.k.a.* $R^2$ score, that yields for each voxel the relative discrepancy between the true time course and the predicted one. For each compared algorithm, the experiment is run 25 times with different seeds to obtain error bars. We report the mean $R^2$ score across voxels in a region of interest (see Appendix F for details) and display the results in Fig 4. The error bars represent a $95\%$ confidence interval. The chance level is given by the $R^2$ score of an algorithm that samples the coefficients of its unmixing matrices and dimension reduction operators from a standardized Gaussian. The median chance level is below $10^{-3}$ on all datasets. ShICA-ML yields the best $R^2$ score in all datasets and for any number of components. ShICA-J yields competitive results with respect to MVICA while being much faster to fit. A popular benchmark especially in the SRM community is the time-segment matching experiment [16]: we include such experiments in Appendix H.2. In appendix G, we give the performance of ShICA-ML, ShICA-J and MVICA in form of a table.

## 7   Conclusion, Future work and Societal impact

We introduced the ShICA model as a principled unifying solution to the problems of shared response modelling and GroupICA. ShICA is able to use both the diversity of Gaussian variances and non-Gaussianity for optimal estimation. We presented two algorithms to fit the model: ShICA-J, a fast algorithm that uses noise diversity, and ShICA-ML, a maximum likelihood approach that can use non-Gaussianity on top of noise diversity. ShICA algorithms come with principled procedures for shared components estimation, as well as adaptation and estimation of noise levels in each view (subject) and component. On simulated data, ShICA clearly outperforms all competing methods in terms of the trade-off between statistical accuracy and computation time. On brain imaging data, ShICA gives more stable decompositions for comparable computation times, and more accurately predicts the data of one subject from the data of other subjects, making it a good candidate to perform transfer learning. Our code is available at `https://github.com/hugorichard/ShICA`. [*]

---

[*]Regarding the ethical aspects of this work, we think this work presents exactly the same issues as any brain imaging analysis method related to ICA.

