**Acknowledgement and funding disclosure**    This work has received funding from the European Union's Horizon 2020 Framework Programme for Research and Innovation under the Specific Grant Agreement No. 945539 (Human Brain Project SGA3), the KARAIB AI chair (ANR-20-CHIA-0025-01), the Grant SLAB ERC-StG-676943 and the BrAIN AI chair (ANR-20-CHIA-0016). PA acknowledges funding by the French government under management of Agence Nationale de la Recherche as part of the "Investissements d'avenir" program, reference ANR19-P3IA-0001 (PRAIRIE 3IA Institute). AH received funding from a CIFAR Fellowship.

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

# A  Proofs and Lemmas

## A.1  Proof of Theorem 1

*Proof.* By hypothesis, the covariances verify $C_{ij} = \mathbb{E}[\mathbf{x}_i \mathbf{x}_j^\top] = A_i(I_p + \delta_{ij}\Sigma_i)A_j^\top = A_i'(I_p + \delta_{ij}\Sigma_i')A_j'^\top$ for all $i, j$. We let $P_i = A_i^{-1}A_i'$. The previous relationship for $j \neq i$ gives $P_i P_j^\top = I_p$. Because there are more than 3 views, there is another integer $k \notin \{i, j\}$, and we have $P_i P_k^\top = P_j P_k^\top = I_p$. This shows that $P_i = P_j$: all these matrices are equal, and we call $P$ their common value. The previous equation also gives $PP^\top = I_p$, so $P$ is orthogonal. We have that $s + n_i$ and $s' + n_i'$ have independent components and $s + n_i = P(s' + n_i')$. Lemma 1 (a direct consequence of classical ICA results [18], Theorem 10) gives $P = \Pi^{-1}\Omega\Pi'$ where $\Pi$ and $\Pi'$ are sign and permutation matrices such that the first $g$ components of $\Pi(s + n_i)$ and $\Pi'(s' + n_i')$ are Gaussian, and $\Omega$ is a block diagonal matrix given by

$$\Omega = \begin{bmatrix} \Omega_g & 0 \\ 0 & I_{p-g} \end{bmatrix}$$

where $\Omega_g$ is orthogonal. We call $A^{(g)}$ the first $g \times g$ block of a matrix $A$ so that $\Omega^{(g)} = \Omega_g$.

Then, considering only the Gaussian components, we can write for $i = j$: $(\Pi\Sigma_i)^{(g)} = \Omega_g(\Pi'\Sigma_i')^{(g)}\Omega_g^\top$ for all $i$. This, combined with Assumption 1, implies that $\Omega_g$ is a sign and permutation matrix (see Lemma 2) and therefore $P$ is a sign and permutation matrix. Then it follows that $I + \Sigma_i = P(I + \Sigma_i')P^\top$ and therefore $\Sigma_i = P\Sigma_i'P^\top$ so $\Sigma_i' = P^\top\Sigma_i P$.  □

## A.2  Proof of Theorem 2

*Proof.* Let us denote $W \in \mathbb{R}^{mp \times mp}$ the block diagonal matrix with block $i$ given by $(A^i)^{-1}$. We have $\mathcal{C}\mathbf{u} = \lambda\mathcal{D}\mathbf{u} \iff W\mathcal{C}W^\top\mathbf{z} = \lambda W\mathcal{D}W^\top\mathbf{z}$ where $\mathbf{u} = W^\top\mathbf{z}$. We call $\mathbf{z}$ a reduced eigenvector.

Each block in $W\mathcal{C}W^\top$ and in $W\mathcal{D}W^\top$ is diagonal so any reduced eigenvector $\mathbf{z} = \begin{bmatrix} \mathbf{z}_1 \\ \vdots \\ \mathbf{z}_m \end{bmatrix}$ is such

that the matrix $Z = [\mathbf{z}_1 \ldots \mathbf{z}_m]$ has exactly one non-zero line. Following Lemma 3, the first $p$ leading reduced eigenvectors $\mathbf{z}^1, \ldots, \mathbf{z}^p$ all have different first non-zero coordinates. Therefore the

concatenation of the first $p$ leading reduced eigenvectors is given by $[\mathbf{z}^1, \ldots \mathbf{z}^p] = \begin{bmatrix} \Gamma_1 \\ \vdots \\ \Gamma_m \end{bmatrix} P^\top$ where

$P^\top \in \mathbb{R}^{p \times p}$ is a permutation matrix and $\Gamma_i \in \mathbb{R}^{p \times p}$ is a diagonal matrix. Therefore, the first $p$

eigenvectors are given by $[\mathbf{u}^1 \ldots \mathbf{u}^p] = \begin{bmatrix} W_1^\top \\ \vdots \\ W_m^\top \end{bmatrix} = \begin{bmatrix} (A_1^{-1})^\top\Gamma_1 P^\top \\ \vdots \\ (A_m^{-1})^\top\Gamma_m P^\top \end{bmatrix}$ and so $W_i = P\Gamma_i A_i^{-1}$  □

**Lemma 1.** *Let $\mathbf{s} \in \mathbb{R}^k$ and $\mathbf{s}' \in \mathbb{R}^k$ have independent components among which $g$ are Gaussian, and $P$ a rotation matrix such that $\mathbf{s} = P\mathbf{s}'$. Then, $P = \Pi^{-1}O\Pi'$ where $\Pi$ and $\Pi'$ are sign and permutation matrices such that the first $g$ components of $\Pi\mathbf{s}$ and $\Pi'\mathbf{s}'$ are Gaussian and $O$ is a block diagonal matrix such that $O^{(g)}$, the first $g \times g$ block of $O$, is orthogonal and the other block is identity.*

*Proof.* From [18], Theorem 10: Assume $\mathbf{s} = P\mathbf{s}'$, if the column $j$ of $P$ has more than one non-zero element then $s_j'$ is Gaussian.

Let us define permutations $\Pi_1, \Pi_1'$ such that the first $g$ components of $\Pi_1\mathbf{s}$ and $\Pi_1'\mathbf{s}'$ are Gaussian and $P_1 = \Pi_1 P(\Pi_1')^{-1}$. We can see that $P_1$ is orthogonal.

We have $\Pi_1\mathbf{s} = P_1\Pi_1'\mathbf{s}'$. So the last $p - g$ columns of $P_1$ contain at most one non-zero element. Using orthogonality of $P_1$ this non-zero element has value 1 or $-1$ and is also the only one in its line. Let us focus on column $l > g$. Assume column $l$ has its non-zero element at index $k \leq g$. Then line $k$ in $P_1$ is only non-zero at index $l$ and therefore $(\Pi_1\mathbf{s})_k$ (which is Gaussian) is equal to $(\Pi_1'\mathbf{s}')_l$ (which is not). Therefore column $l$ can only have its non-zero element at an index greater than $g$.

This shows that $P_1$ is block diagonal $P_1 = \begin{bmatrix} O_g & 0 \\ 0 & P_2 \end{bmatrix}$ where $O_g$ is orthogonal and $P_2$ is a sign and permutation matrix.

$$\begin{bmatrix} O_g & 0 \\ 0 & P_2 \end{bmatrix} = \Pi_1 P (\Pi_1')^{-1} \tag{7}$$

$$\iff \begin{bmatrix} O_g & 0 \\ 0 & I \end{bmatrix} \begin{bmatrix} I & 0 \\ 0 & P_2 \end{bmatrix} = \Pi_1 P (\Pi_1')^{-1} \tag{8}$$

$$\iff \Pi_1^{-1} \begin{bmatrix} O_g & 0 \\ 0 & I \end{bmatrix} \begin{bmatrix} I & 0 \\ 0 & P_2 \end{bmatrix} \Pi_1' = P \tag{9}$$

Therefore setting $\Pi' = \begin{bmatrix} I & 0 \\ 0 & P_2 \end{bmatrix} \Pi_1'$ and $\Pi = \Pi_1$ and $O = \begin{bmatrix} O_g & 0 \\ 0 & I \end{bmatrix}$ concludes the proof.

$\square$

**Lemma 2.** *Assume that Assumption 2 holds for $\Sigma_i$, and that there is an orthogonal matrix $P$ and diagonal matrices $\Sigma_i'$ such that for all $i$, $\Sigma_i' = P\Sigma_i P^\top$. Then, $P$ is a permutation matrix.*

*Proof.* The proof is in two parts. First, we show that there exist some coefficients $\alpha_1, \ldots, \alpha_m$ such that the matrix $\sum_i \alpha_i \Sigma_i$ has distinct coefficients on the diagonal. Then, since we have $\sum_i \alpha_i \Sigma_i' = P \left( \sum_i \alpha_i \Sigma_i \right) P^\top$, and the diagonal $\sum_i \alpha_i \Sigma_i$ has distinct entries, we can invoke the unicity of the eigenvalue decomposition for symmetric matrices, which shows that $P$ is necessarily a permutation matrix. Now, the only thing left is to prove is that Assumption 2 implies the existence of this linear combination.

We assume by contradiction that any linear combination of the $\Sigma_i$ has two equal entries.

For $\alpha = [\alpha_1, \ldots, \alpha_m]$, we let $\mathcal{S}(\alpha) = \text{diag}(\sum_i \alpha_i \Sigma_i) \in \mathbb{R}^p$, where $\text{diag}(\cdot)$ extracts the diagonal entries. The operator $\mathcal{S}$ is linear. We now define for $j, j' \leq p$ the linear form $\ell_{jj'}(\alpha) = \mathcal{S}(\alpha)_j - \mathcal{S}(\alpha)_{j'} \in \mathbb{R}$. The assumption on the linear combinations of $\Sigma_i$ simply rewrites: For all $\alpha \in \mathbb{R}^m$, there exists $j, j' \leq p$ such that $\ell_{jj'}(\alpha) = 0$.

From a set point of view, this relationship writes

$$\bigcup_{j,j'} \text{Ker}(\ell_{jj'}) = \mathbb{R}^m \ .$$

Since the $\ell_{jj'}$ are all linear forms, the $\text{Ker}(\ell_{jj'})$ are subspaces of dimensions $m$ or $m-1$, and since their union is of dimension $m$, there exists $j, j'$ such that $\text{Ker}(\ell_{jj'}) = \mathbb{R}^m$, i.e. such that $\ell_{jj'} = 0$.

As a consequence, we have for all $\alpha$, $\mathcal{S}(\alpha)_j = \mathcal{S}(\alpha)_{j'}$. This implies that the sequences $(\Sigma_{ij})_i$ and $(\Sigma_{ij'})_i$ are equal, which contradicts Assumption 2.

We have therefore shown that Assumption 2 implies the existence of a linear combination of the $\Sigma_i$ that has distinct entries, which concludes the proof. $\square$

**Lemma 3.** *Let us consider the following eigenvalue problem:*

$$\begin{bmatrix} I + \Sigma_1 & I & \ldots & I \\ I & I + \Sigma_2 & \ddots & \vdots \\ \vdots & \ddots & \ddots & I \\ I & \ldots & I & I + \Sigma_m \end{bmatrix} \mathbf{z} = \lambda \begin{bmatrix} I + \Sigma_1 & 0 & \ldots & 0 \\ 0 & I + \Sigma_2 & \ddots & \vdots \\ \vdots & \ddots & \ddots & 0 \\ 0 & \ldots & 0 & I + \Sigma_m \end{bmatrix} \mathbf{z} \tag{10}$$

*where $\forall i, \ 1 \leq i \leq m, \ \Sigma_m \in \mathbb{R}^{p,p}$ are positive diagonal matrices and $I$ is the identity matrix. If the first $p$ eigenvalues are distincts, the first $p$ eigenvectors $\mathbf{z}^1, \ldots, \mathbf{z}^p, \mathbf{z}^i \in \mathbb{R}^{mp}$ have different first non-zero coordinates.*

*Proof.* We sort the eigenvectors in $p$ groups of $m$ vectors so that all vectors in group $l$ have their $l$-th coordinate different from 0. Let $\mathbf{z}^{(l)}$ be an eigenvector in group $l$ and let us call $\mathbf{w}_l \in \mathbb{R}^m$ the non-zero coordinates of this eigenvector: $\forall i \in \{1 \ldots m\}, w_{li} = z_{l+(i-1)p}^{(l)}$.

We have:

$$
\begin{bmatrix}
1+\Sigma_{1l} & 1 & \dots & 1 \\
1 & 1+\Sigma_{2l} & \ddots & \vdots \\
\vdots & \ddots & \ddots & 1 \\
1 & \dots & 1 & 1+\Sigma_{ml}
\end{bmatrix}
\mathbf{w}_l =
\begin{bmatrix}
1+\Sigma_{1l} & 0 & \dots & 0 \\
0 & 1+\Sigma_{2l} & \ddots & \vdots \\
\vdots & \ddots & \ddots & 0 \\
0 & \dots & 0 & 1+\Sigma_{ml}
\end{bmatrix}
\mathbf{w}_l \lambda_l \quad (11)
$$

We now show that the biggest eigenvalue of (11) is strictly above 1 while all others are strictly below 1. The core of the proof comes from the study of the eigenvalues of a matrix modified by a rank 1 matrix. The reasoning we use here follows [25] (end of section 5).

Let us introduce $K^l = \mathrm{diag}(\Sigma_{1l} \dots \Sigma_{ml})$ and $\mathbf{u} = \begin{bmatrix} 1 \\ \vdots \\ 1 \end{bmatrix}$. Let us drop the index $l$ in the notations for simplicity.

The problem can be rewritten

$$
(\mathbf{u}\mathbf{u}^\top + K)\mathbf{w} = (I + K)\mathbf{w}\lambda \tag{12}
$$

$$
\iff (I+K)^{-1}(\mathbf{u}\mathbf{u}^\top + K)\mathbf{w} = \mathbf{w}\lambda \tag{13}
$$

The characteristic polynomial is given by:

$$
\mathcal{P}(\lambda) = \det((I+K)^{-1}K - \lambda I + (I+K)^{-1}\mathbf{u}\mathbf{u}^\top) \tag{14}
$$

$$
\propto \det(I + ((I+K)^{-1}K - \lambda I)^{-1}(I+K)^{-1}\mathbf{u}\mathbf{u}^\top) \tag{15}
$$

where we implicitly focus here on eigenvalues $\lambda$ such that $\det((I+K)^{-1}K - \lambda I) \neq 0 \iff \forall i, \lambda \neq \frac{k_i}{1+k_i}$.

We then use the following property: Let $A \in \mathbb{R}^{a,b}$ and $B \in \mathbb{R}^{b,a}$ we have $\det(I_a + AB) = \det(I_b + BA)$.

Let us call $\chi(\lambda) = \det(I + ((I+K)^{-1}K - \lambda I)^{-1}(I+K)^{-1}\mathbf{u}\mathbf{u}^\top)$ we have:

$$
\chi(\lambda) = 1 + \mathbf{u}^\top((I+K)^{-1}K - \lambda I)^{-1}(I+K)^{-1}\mathbf{u} \tag{16}
$$

$$
= 1 + \sum_{i=1}^{m} \frac{1}{1+k_i} \frac{1}{\frac{k_i}{1+k_i} - \lambda} \tag{17}
$$

where $k_i = \Sigma_{il} > 0$. Taking the derivative we get

$$
\chi'(\lambda) = \sum_{i=1}^{m} \frac{1}{1+k_i} \frac{1}{(\frac{k_i}{1+k_i} - \lambda)^2} > 0 \tag{18}
$$

Trivially, $\forall i, \frac{k_i}{1+k_i} < 1$. We also have

$$
\chi(1) = 1 + \sum_{i=1}^{m} \frac{1}{1+k_i} \frac{1}{\frac{k_i}{1+k_i} - 1} = 1 - m < 0 \tag{19}
$$

and $\lim_{\lambda \to +\infty} \chi(\lambda) = 1$ so as $\chi$ is continuous and strictly increasing on $[1, +\infty[$. Therefore, it reaches 0 only once on this interval (excluding 1 since we know $\chi(1) \neq 0$). Therefore the greatest eigenvalue $\lambda^*$ is strictly above 1 while all other eigenvalues are strictly below 1.

Note that because $\chi' > 0$, $\lambda^*$ is of multiplicity 1. In the analysis above we ignored those eigenvalues $\lambda$ such that $\lambda = \frac{k_i}{1+k_i}$ for some $i$. However since $\frac{k_i}{1+k_i} < 1$, none of these eigenvalues can be the largest one.

Finally, the $p$ first eigenvectors belong to different groups (the corresponding eigenvalues are all strictly above 1). This shows that these eigenvectors have different first non-zero coordinates.

$\square$

## B  Identifiability results for $m < 3$

We have a slightly weaker identifiability result when $m = 2$.

**Proposition 5.** *Let $m = 2$, and suppose that the scalars $(1 + \Sigma_{1j})(1 + \Sigma_{2j})$ for $j = 1 \dots p$ are all different. We let $\Theta' = (A'_1, A'_2, \Sigma'_1, \Sigma'_2)$ that also generates $\mathbf{x}_1, \mathbf{x}_2$. Then, there exists a permutation and scale matrix $P$ such that $A'_1 = A_1 P$ and $A'_2 = A_2 P^{-\top}$.*

*Proof.* We let $P = A_1^{-1} A'_1$. Since $C_{12} = I_p$, it holds $A_2^{-1} A'_2 = P^{-\top}$. Then, we have $I_p + \Sigma_1 = P(I_p + \Sigma'_1)P^\top$. This means that there exists $U \in \mathcal{O}_p$ such that $P = (I_p + \Sigma_1)^{\frac{1}{2}} U (I_p + \Sigma'_1)^{-\frac{1}{2}}$. Since $P^{-\top}(I_p + \Sigma'_2)P^{-1} = I_p + \Sigma_2$, we find $U(I_p + \Sigma'_1)(I_p + \Sigma'_2)U^\top = (I_p + \Sigma_1)(I_p + \Sigma_2)$. By identification, $U$ is a permutation matrix, and $P$ is a scale and permutation matrix. □

As a consequence, when there are only two subjects, it is possible to recover the components and noise levels up to a scaling factor. When there is only one view, $m = 1$, there is a global rotation indeterminacy: $A_1(I_p + \Sigma_1)A_1^\top = A'_1(I_p + \Sigma_1)A'_1{}^\top$ for $A'_1 = A_1(I_p + \Sigma_1)^{\frac{1}{2}} U(I_p + \Sigma_1)^{-\frac{1}{2}}$ where $U$ is any orthogonal matrix. In this case, we lose identifiability.

## C  Derivation of fixed point updates for scalings

We want to minimize

$$L((\Phi_i)_{i=1}^m) = \sum_i \sum_{j \neq i} \|\Phi_i \mathrm{diag}(Y_{ij})\Phi_j - I_p\|_F^2 \tag{20}$$

for $\Phi_i$ diagonal. With respect to each $\Phi_i$, this function is strongly-convex, which means that the minimization w.r.t $\Phi_i$ can be done by cancelling the gradient. The gradient is given by

$$\frac{\partial L}{\partial \Phi_i} = 2 \sum_{j \neq i} (\Phi_i \mathrm{diag}(Y_{ij})\Phi_j - I_p)\Phi_j \tag{21}$$

Therefore we get

$$\frac{\partial L}{\partial \Phi_i} = 0 \tag{22}$$

$$\iff 2 \sum_{j \neq i} (\Phi_i \mathrm{diag}(Y_{ij})\Phi_j - I_p)\Phi_j = 0 \tag{23}$$

$$\iff \Phi_i \sum_{j \neq i} \mathrm{diag}(Y_{ij})\Phi_j^2 - \sum_{j \neq i} \Phi_j = 0 \tag{24}$$

$$\iff \Phi_i = \frac{\sum_{j \neq i} \Phi_j}{\sum_{j \neq i} \mathrm{diag}(Y_{ij})\Phi_j^2} \tag{25}$$

This update equation ensures that $\Phi_i = \arg\min_{\Phi_i} L((\Phi_j)_{i=1}^m)$, and we then loop through the $\Phi_i$ to get an alternate minimization scheme, which is guaranteed to converge to a stationary point of (20).

## D  EM E-step and M-step for ShICA with Gaussian components

### D.1  E-step

The derivations are the same as in section E.1 but the sum over $\alpha \in \frac{1}{2}, \frac{3}{2}$ is replaced by just $\alpha = 1$.

## D.2 M-step

The function to minimize in the M-step is then given by:

$$\mathcal{J} = -\log p(\mathbf{x}, \mathbf{s}) \tag{26}$$

$$= \sum_{i=1}^{m} \log(|\Sigma_i|) + \frac{1}{2} \operatorname{tr}(\Sigma_i^{-1} \left[ (\mathbf{y}_i - \mathbb{E}[\mathbf{s}|\mathbf{x}])(\mathbf{y}_i - \mathbb{E}[\mathbf{s}|\mathbf{x}])^\top + \mathbb{V}[\mathbf{s}|\mathbf{x}] \right]) + c \tag{27}$$

where $c$ does not depend on $\Sigma_i$

Therefore we get closed-form updates for $\Sigma_i$:

$$\Sigma_i \leftarrow \operatorname{diag}((\mathbf{y}_i - \mathbb{E}[\mathbf{s}|\mathbf{x}])(\mathbf{y}_i - \mathbb{E}[\mathbf{s}|\mathbf{x}])^\top + \mathbb{V}[\mathbf{s}|\mathbf{x}]) \tag{28}$$

Plugging in the closed-form formula for $\mathbb{E}[\mathbf{s}|\mathbf{x}]$ and $\mathbb{V}[\mathbf{s}|\mathbf{x}]$ we get updates that only depends on the covariances $\hat{C}_{ij} = \mathbb{E}[\mathbf{x}_i \mathbf{x}_j^\top]$.

$$\Sigma_i \leftarrow \operatorname{diag}(\hat{C}_{ii} - 2\mathbb{V}[\mathbf{s}|\mathbf{x}] \sum_{j=1}^{m} \Sigma_j^{-1} \hat{C}_{ji} + \mathbb{V}[\mathbf{s}|\mathbf{x}] \sum_{j=1}^{m} \sum_{l=1}^{m} \left( \Sigma_j^{-1} \hat{C}_{jl} \Sigma_l^{-1} \right) \mathbb{V}[\mathbf{s}|\mathbf{x}] + \mathbb{V}[\mathbf{s}|\mathbf{x}])$$

# E EM E-step and M-step for ShICA with non-Gaussian components

## E.1 E-step

The complete likelihood is given by

$$p(\mathbf{x}, \mathbf{s}) = \prod_i p(\mathbf{x}_i|\mathbf{s})p(\mathbf{s}) \tag{29}$$

$$= \prod_i p(\mathbf{x}_i|\mathbf{s}) \prod_j \sum_{\alpha \in \{0.5, 1.5\}} p(s_j|\alpha) \tag{30}$$

$$\tag{31}$$

where

$$p(s_j|\alpha) = \mathcal{N}(s_j; 0, \alpha) \tag{32}$$

We have

$$p(\mathbf{x}_i|\mathbf{s}) = |W_i| \mathcal{N}(\mathbf{y}_i; \mathbf{s}, \Sigma_i) \tag{33}$$

$$= |W_i| \prod_j \mathcal{N}(y_{ij}; s_j, \Sigma_{ij}) \tag{34}$$

where $\Sigma_{ij}$ is the coefficient $j, j$ of $\Sigma_i$ and $\mathbf{y}_i = W\mathbf{x}_i$.

Let us introduce a first lemma:

**Lemma 4.**

$$\prod_{i=1}^{m} \mathcal{N}(x_i; u, v_i) = \prod_{i=1}^{m} \mathcal{N}(x_i; \bar{x}, v_i) \sqrt{2\pi\bar{v}} \mathcal{N}(\bar{x}; u, \bar{v})$$

where $\bar{v} = (\sum_{i=1}^{m} v_i^{-1})^{-1}$ and $\bar{x} = \frac{\sum_i v_i^{-1} x_i}{\sum_i v_i^{-1}}$.

*Proof.* We have that

$$\sum_i \frac{1}{v_i}(x_i - u)^2 = \sum_i \frac{1}{v_i}(x_i - u)^2 \tag{35}$$

$$= \sum_i \frac{1}{v_i}(x_i - \bar{x} + \bar{x} - u)^2 \tag{36}$$

$$= \sum_i \frac{1}{v_i}(x_i - \bar{x})^2 + \sum_i \frac{1}{v_i}(\bar{x} - u)^2 \tag{37}$$

and therefore

$$\prod_i (\frac{1}{\sqrt{2\pi v_i}} \exp(-\frac{1}{2v_i}(x_i - \mu)^2)) \tag{38}$$

$$= \prod_i \frac{1}{\sqrt{2\pi v_i}} \exp(\sum_i -\frac{1}{2}(\frac{1}{v_i}(x_i - \bar{x})^2 + \frac{1}{v_i}(\bar{x} - u)^2)) \tag{39}$$

$$= \prod_i \mathcal{N}(x_i, \bar{x}, v_i) \exp(-\frac{1}{2}(\sum_i \frac{1}{v_i})(\bar{x} - u)^2)) \tag{40}$$

$$\tag{41}$$

so the desired result follow. $\qquad \square$

By Lemma 4, we have

$$\prod_i p(\mathbf{x}_i | \mathbf{s}) = \prod_i |W_i| \prod_j \mathcal{N}(y_{ij}; \bar{y}_j, \Sigma_{ij}) \sqrt{2\pi \bar{\Sigma}_j} \mathcal{N}(\bar{y}_j; s_j, \bar{\Sigma}_j) \tag{42}$$

$$\tag{43}$$

where $\bar{y}_j = \frac{\sum_i \Sigma_{ij}^{-1} y_{ij}}{\sum_i \Sigma_{ij}^{-1}}$ and $\bar{\Sigma}_j = (\sum_i \Sigma_{ij}^{-1})^{-1}$. Hiding variable that do not depend on $\mathbf{s}$ we obtain

$$\prod_i p(\mathbf{x}_i | \mathbf{s}) \propto \prod_j \mathcal{N}(\bar{y}_j; s_j, \bar{\Sigma}_j) \tag{44}$$

$$\tag{45}$$

Then we get

$$p(\mathbf{x}, \mathbf{s}) \propto \prod_j \sum_{\alpha \in \{0.5, 1.5\}} \mathcal{N}(s_j; \bar{y}_j, \bar{\Sigma}_j) \mathcal{N}(s_j; 0, \alpha) \tag{46}$$

Let us now prove a second Lemma:

**Lemma 5.**

$$\mathcal{N}(x; y, \nu) \mathcal{N}(x, 0, \alpha) = \mathcal{N}(y; 0, \nu + \alpha) \mathcal{N}(x; \frac{\alpha y}{\alpha + \nu}, \frac{\nu \alpha}{\alpha + \nu})$$

*Proof.* We have

$$\mathcal{N}(x; y, \nu) \mathcal{N}(x, 0, \alpha) = \frac{\exp\left(-\frac{(x-y)^2}{2\nu}\right)}{\sqrt{2\pi\nu}} \frac{\exp\left(-\frac{x^2}{2\alpha}\right)}{\sqrt{2\pi\alpha}} \tag{47}$$

Then,

$$\exp\left(-\frac{(x-y)^2}{2\nu}\right) \tag{48}$$

$$= \exp\left(-\frac{\alpha(x-y)^2 + \nu x^2}{2\alpha\nu}\right) \tag{49}$$

$$= \exp\left(-\frac{\alpha(x^2 - 2xy + y^2) + \nu x^2}{2\alpha\nu}\right) \tag{50}$$

$$= \exp\left(-\frac{x^2(\alpha+\nu) - 2x(\alpha y) + \alpha y^2}{2\alpha\nu}\right) \tag{51}$$

$$= \exp\left(-\frac{x^2 - 2x\frac{\alpha y}{\alpha+\nu} + \frac{\alpha y^2}{\alpha+\nu}}{2\frac{\alpha\nu}{\alpha+\nu}}\right) \tag{52}$$

$$= \exp\left(-\frac{(x - \frac{\alpha y}{\alpha+\nu})^2 - (\frac{\alpha y}{\alpha+\nu})^2 + \frac{\alpha y^2}{\alpha+\nu}}{2\frac{\alpha\nu}{\alpha+\nu}}\right) \tag{53}$$

$$= \exp\left(-\frac{(x - \frac{\alpha y}{\alpha+\nu})^2}{2\frac{\alpha\nu}{\alpha+\nu}}\right)\exp\left(-\frac{-\alpha^2 y^2 + (\alpha+\nu)\alpha y^2}{2\alpha\nu(\alpha+\nu)}\right) \tag{54}$$

$$= \exp\left(-\frac{(x - \frac{\alpha y}{\alpha+\nu})^2}{2\frac{\alpha\nu}{\alpha+\nu}}\right)\exp\left(-\frac{\nu\alpha y^2}{2\alpha\nu(\alpha+\nu)}\right) \tag{55}$$

and

$$\frac{1}{\sqrt{2\pi\nu}\sqrt{2\pi\alpha}} = \frac{1}{\sqrt{2\pi(\nu+\alpha)}\sqrt{2\pi\frac{\nu\alpha}{\nu+\alpha}}} \tag{56}$$

so that the desired result follow. $\qquad\square$

By Lemma 5, we have:

$$p(\mathbf{x}, \mathbf{s}) \tag{57}$$

$$\propto \prod_j \sum_{\alpha\in\{0.5,1.5\}} \mathcal{N}(\bar{y}_j; 0, \bar{\Sigma}_j + \alpha)\mathcal{N}(s_j; \frac{\alpha\bar{y}_j}{\alpha+\bar{\Sigma}_j}, \frac{\bar{\Sigma}_j\alpha}{\alpha+\bar{\Sigma}_j}) \tag{58}$$

and therefore we get:

$$p(\mathbf{s}|\mathbf{x}) = \frac{p(\mathbf{s}, \mathbf{x})}{\int_{\mathbf{s}} p(\mathbf{s}, \mathbf{x})} \tag{59}$$

$$= \prod_j \frac{\sum_{\alpha\in\{0.5,1.5\}} \theta_\alpha \mathcal{N}(s_j; \frac{\alpha\bar{y}_j}{\alpha+\bar{\Sigma}_j}, \frac{\bar{\Sigma}_j\alpha}{\alpha+\bar{\Sigma}_j})}{\sum_{\alpha\in\{0.5,1.5\}} \theta_\alpha} \tag{60}$$

where $\theta_\alpha = \mathcal{N}(\bar{y}_j; 0, \bar{\Sigma}_j + \alpha)$.

So we obtain the desired result:

$$\mathbb{E}[s_j|\mathbf{x}] = \frac{\sum_{\alpha\in\{0.5,1.5\}} \theta_\alpha \frac{\alpha\bar{y}_j}{\alpha+\bar{\Sigma}_j}}{\sum_{\alpha\in\{0.5,1.5\}} \theta_\alpha} \tag{61}$$

$$\mathbb{V}[s_j|\mathbf{x}] = \frac{\sum_{\alpha\in\{0.5,1.5\}} \theta_\alpha \frac{\bar{\Sigma}_j\alpha}{\alpha+\bar{\Sigma}_j}}{\sum_{\alpha\in\{0.5,1.5\}} \theta_\alpha} \tag{62}$$

## E.2 M-step

The function to minimize in the M-step is then given by:

$$\mathcal{J} = -\log p(\mathbf{x}, \mathbf{s}) \tag{63}$$

$$= \sum_{i=1}^{m} -\log(|W_i|) + \log(|\Sigma_i|) + \frac{1}{2} \operatorname{tr}(\Sigma_i^{-1} \left[ (\mathbf{y}_i - \mathbb{E}[\mathbf{s}|\mathbf{x}])(\mathbf{y}_i - \mathbb{E}[\mathbf{s}|\mathbf{x}])^\top + \mathbb{V}[\mathbf{s}|\mathbf{x}] \right]) + c \tag{64}$$

where $c$ does not depend on $\Sigma_i$ or $W_i$

Therefore we get closed-form updates for $\Sigma_i$:

$$\Sigma_i \leftarrow \operatorname{diag}((\mathbf{y}_i - \mathbb{E}[\mathbf{s}|\mathbf{x}])(\mathbf{y}_i - \mathbb{E}[\mathbf{s}|\mathbf{x}])^\top + \mathbb{V}[\mathbf{s}|\mathbf{x}]) \tag{65}$$

We update $W_i$ by performing a quasi-Newton step.

We use the relative gradient $\mathcal{G}^{W_i}$ and $\mathcal{H}^{W_i}$ defined by
$\mathcal{J}(W_i + \varepsilon W_i) = \mathcal{J}(W_i) + \langle \varepsilon | \mathcal{G}^{W_i} \rangle + \frac{1}{2} \langle \varepsilon | \mathcal{H}^{W_i} \varepsilon \rangle$.

We get:

$$\mathcal{J}(W_i + \varepsilon W_i) = \sum_{i=1}^{m} \left[ -\log(|W_i|) - \log(|I_k + \varepsilon|) - \log(\mathcal{N}(\mathbf{y}_i + \varepsilon \mathbf{y}^i; \mathbf{s}; \Sigma_i)) \right] + const \tag{66}$$

$$= \mathcal{J}(W_i) - \operatorname{tr}(\varepsilon) + \frac{1}{2} \operatorname{tr}(\varepsilon^2) \tag{67}$$

$$+ \frac{1}{2} \left[ \langle \varepsilon \mathbf{y}^i | (\Sigma_i)^{-1} (\mathbf{y}^i - \mathbf{s}) \rangle + \langle (\mathbf{y}^i - \mathbf{s}) | (\Sigma_i)^{-1} \varepsilon \mathbf{y}^i \rangle + \langle \varepsilon \mathbf{y}^i | (\Sigma_i)^{-1} \varepsilon \mathbf{y}^i \rangle \right] \tag{68}$$

$$+ o(\|\varepsilon\|^2) \tag{69}$$

$$= \mathcal{J}(W_i) - \sum_{a} \varepsilon_{a,a} + \frac{1}{2} \sum_{a,b} \varepsilon_{a,b} \varepsilon_{b,a} \tag{70}$$

$$+ \sum_{a,b} \varepsilon_{a,b} \left[ (\Sigma_i)^{-1} (\mathbf{y}^i - \mathbf{s})(\mathbf{y}^i)^\top \right]_{a,b} + \frac{1}{2} \sum_{a,b} \varepsilon_{a,b} \left[ (\Sigma_i)^{-1} \varepsilon \mathbf{y}^i (\mathbf{y}^i)^\top \right]_{a,b} \tag{71}$$

$$+ o(\|\varepsilon\|^2) \tag{72}$$

$$= \mathcal{J}(W_i) - \sum_{a} \varepsilon_{a,a} + \frac{1}{2} \sum_{a,b} \varepsilon_{a,b} \varepsilon_{b,a} \tag{73}$$

$$+ \sum_{a,b} \varepsilon_{a,b} \left[ (\Sigma_i)^{-1} (\mathbf{y}^i - \mathbf{s})(\mathbf{y}^i)^\top \right]_{a,b} + \frac{1}{2} \sum_{a,b,d} \varepsilon_{a,b} (\Sigma_i)_{a,a}^{-1} \varepsilon_{a,d} \left[ \mathbf{y}^i (\mathbf{y}^i)^\top \right]_{d,b} \tag{74}$$

$$+ o(\|\varepsilon\|^2) \tag{75}$$

$$\tag{76}$$

So:

$$\mathcal{G}_{a,b}^{W_i} = -\delta_{a,b} + \left[ (\Sigma_i)^{-1} (\mathbf{y}^i - \mathbf{s})(\mathbf{y}^i)^\top \right]_{a,b} \tag{77}$$

and

$$\mathcal{H}_{a,b,c,d}^{W_i} = \delta_{a,d} \delta_{b,c} + \delta_{a,c} \frac{y_{ib} y_{id}}{\Sigma_{ia}} \tag{78}$$

We approximate the Hessian by

$$\widehat{\mathcal{H}_{a,b,c,d}^{W_i}} = \delta_{ad} \delta_{bc} + \delta_{ac} \delta_{bd} \frac{(y_{ib})^2}{\Sigma_{ia}} \tag{79}$$

where the Hessian approximation is exact when the unmixed data have truly independent components.

Updates for $W_i$ are then given by $W_i \leftarrow (I - \rho(\widehat{\mathcal{H}^{W_i}})^{-1} \mathcal{G}^{W_i}) W_i$, where $\rho$ is chosen by backtracking line-search. We alternate between computing the statistics $\mathbb{E}[\mathbf{s}|\mathbf{x}]$ and $\operatorname{Var}[\mathbf{s}|\mathbf{x}]$ (E-step) and updates of parameters $\Sigma_i$ and $W_i$ for $i = 1 \ldots m$ (M-step).

| Dataset | Duration | $m$ | Description |
|---|---|---|---|
| Sherlock | 50 min | 16 | Movie watching (BBC TV show "Sherlock") |
| Forrest | 110 min | 19 | Auditory version "Forrest Gump" |
| Gallant | 130 min | 12 | various short video clips |
| Raiders | 110 min | 11 | Movie watching ("Raiders of the lost ark") |

Table 1: Information about datasets (name, duration, number of subjects $m$ and short description)

## F   Description of the datasets and the preprocessing pipeline

All datasets are resampled and masked using the brain mask available at `http://cogspaces.github.io/assets/data/hcp_mask.nii.gz`. The dimensionality of the data is given by the number of voxels in the mask: 212445. Data are detrended and standardized so that each voxels' timecourse has zero mean and unit variance.

When reconstructing the BOLD signal of missing subjects, data are preprocessed with a 6 mm smoothing. In the timesegment matching experiment, we use unsmoothed data except for the sherlock dataset for which the available data are already smoothed. Multiple acquisitions (called runs) are necessary to build the datasets. Each run lasts approximately 10 minutes.

Sherlock data are available at `http://arks.princeton.edu/ark:/88435/dsp01nz8062179`. We refer the reader to [15] for a precise description of the study cohort, experimental design and pre-processing pipeline. The data are split manually into 4 runs of 395 timeframes and one run of 396 timeframes so that cross validation can be performed. Subject 5 is removed because of missing data. The repetition time (TR) is 1.5s and the spatial resolution is of 3 mm.

Forrest data are downloaded from OpenfMRI [50]. Data are acquired with a 7T scanner with an isotropic spatial resolution of 1 mm and then resampled to a spatial resolution of 3 mm. A complete description of the experimental design and study cohort are given in `http://studyforrest.org` and [28]. Subject 10 is discarded as not all runs are available at the time of writing. Run 8 is discarded as it is missing in some subjects. We therefore uses 7 runs of respectively 451, 441, 438, 488, 462, 439 and 542 timeframes and 19 subjects. The repetition time (TR) is 2s and the spatial resolution is of 1 mm.

Raiders and Gallant dataset pertains to the Individual Brain Charting dataset. These data were acquired using a 3T scanner and resampled to an isotropic spatial resolution of 3 mm. More information is available in [49]. Gallant dataset is refered to as clips in [49]. Data are available at `https://openneuro.org/datasets/ds00268`. Datasets gallant and raiders are preprocessed using FSL `http://fsl.fmrib.ox.ac.uk/fsl` using slice time correction, spatial realignment, co-registration to the T1 image and affine transformation of the functional volumes to a template brain (MNI). The repetition time (TR) is 2s and the spatial resolution is of 3 mm. The Raiders dataset uses 9 runs of respectively 374, 297, 314, 379, 347, 346, 350, 353 and 211 timeframes. The Gallant dataset uses 17 runs of 325 timeframes each. The protocol used for Raiders is the same as the one used in [30] and the protocol used for Gallant is the same as the one used in [43].

A brief summary of the characteristics of the datasets is available in Table 1

All datasets used in MEG have dimensionality 102 since we only consider the magnetometers. The temporal resolution is 1 ms.

The *CamCAN* dataset [56] contains the MEG data of 496 different subjects exposed to an audio-visual stimuli. More precisely, subjects are presented simultaneously an auditory stimuli lasting 300ms at frequency 300, 600 or 1200 Hz and a checkerboard pattern lasting 34ms. 120 trials are available. The protocol used in the CamCAN MEG dataset is described in [56].

## G   Reconstructing the BOLD signal of missing subjects

We report in Table 2 the R2 score obtained with MVICA, ShICA-J and ShICA-ML with 20 components as well as a 95% confidence interval on the experiment "Reconstructing the fMRI data of left-out subjects". These data are already reported in Figure 4 but are given here in form of a table.

| Dataset | Method | $R^2$ score | Confidence interval |
|---------|--------|-------------|---------------------|
| forrest | ShICA-ML | 0.200 | [0.187, 0.213] |
|         | ShICA-J | 0.171 | [0.157, 0.185] |
|         | MVICA | 0.191 | [0.177, 0.204] |
| gallant | ShICA-ML | 0.121 | [0.107, 0.135] |
|         | ShICA-J | 0.110 | [0.095, 0.125] |
|         | MVICA | 0.114 | [0.099, 0.128] |
| raiders | ShICA-ML | 0.158 | [0.142, 0.174] |
|         | ShICA-J | 0.146 | [0.129, 0.162] |
|         | MVICA | 0.144 | [0.124, 0.164] |
| sherlock | ShICA-ML | 0.174 | [0.157, 0.191] |
|          | ShICA-J | 0.165 | [0.146, 0.183] |
|          | MVICA | 0.161 | [0.142, 0.180] |

Table 2: **Reconstructing the BOLD signal of missing subjects**. Median $R^2$ score and 95% confidence interval.

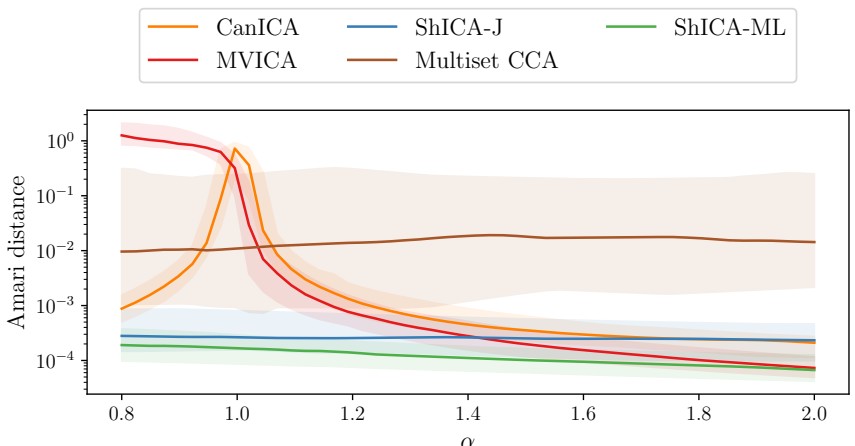

Figure 5: **Separation performance in function of non-Gaussianity** Separation performance of algorithms for sub-Gaussian $\alpha < 1$ and super-Gaussian $\alpha > 1$ components

## H  Additional experiments

### H.1  Separation performance

#### H.1.1  Separation performance in function of non-Gaussianity

We generate data according to model (1). Components s are generated using $s_j = d(x)$ with $d(x) = x|x|^{\alpha-1}$ and $x \sim \mathcal{N}(0,1)$. Mixing matrices $A_i$ are generated by sampling their coefficients from a standardized Gaussian law. The number of samples is fixed to $n = 10^5$ and we vary $\alpha$ between 0.8 and 1.2. Each experiment is repeated 40 times using different seeds in the random number generator. We use $p = 4$ components and $m = 5$ views. We display in Fig 5 the mean Amari distance across subjects. The experiment is repeated 100 times using different seeds. We report the median result and error bars represent the first and last deciles. When $\alpha$ is close to 1 (components are almost Gaussian), ShICA-J, ShICA-ML and multiset CCA can separate components well (but multiset CCA reaches higher amari distance than ShICA). In this regime, MVICA yields much higher amari distance than ShICA-J, ShICA-ML or Multiset CCA but is still better than CanICA which cannot separate components at all. As non-Gaussianity ($\alpha$) increases, ICA based methods yield better results but ShICA-ML yields uniformly lower amari distance.

### H.2  fMRI timesegment matching experiment

We benchmark ShICA on four different real fMRI datasets via a timesegment matching experiment similar to the one in [16]. We use full brain data. The datasets and the preprocessing pipeline are

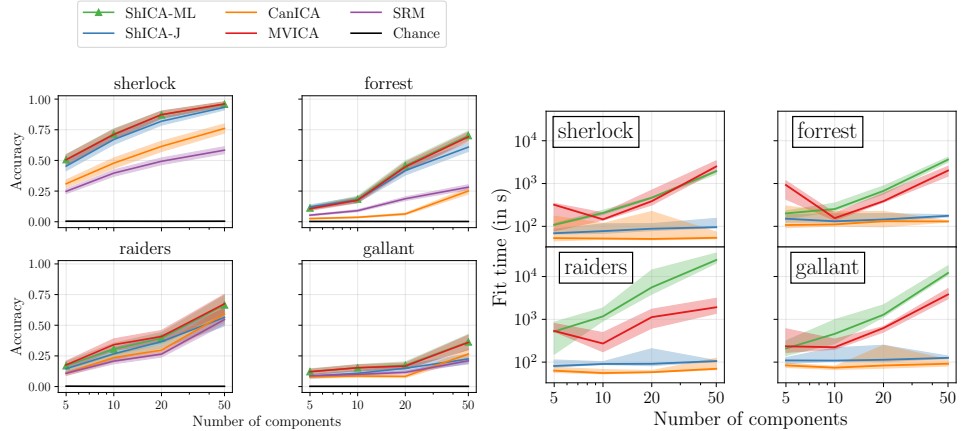

Figure 6: **Timesegment matching experiment**: (left) Accuracy (right) Fitting time (in seconds)

described in Appendix F. We split the data into a train and test set and algorithms are fitted on the train set. On the test set, we estimate the shared components from all subjects but one and select a target timesegment containing 9 consecutive samples in the shared components. We try to localize this timesegment from the components of the left-out subject using a maximum correlation classifier (all possible windows of 9 consecutive timeframes are considered in the left-out subject excluding the ones partially overlapping with the correct timesegment). The left panel in Fig 6 shows that ShICA-ML, MVICA and ShICA-J yield almost equal accuracy and outperform other methods by a large margin. The right panel in Fig 6 shows that ShICA-J is much faster to fit than MVICA or ShICA-ML.

We would like to highlight here that these experiments are not exactly the same as in [16] as we use full brain data and they use regions of interest. The code used for this experiment is very similar to the tutorial in `https://brainiak.org/tutorials/11-SRM/`. We use the SRM implementation in Brainiak [35]. Also note that the Raiders dataset is different from the one used in [16] as it involves different subjects and data were acquired in a different neuro-imaging center.

### H.3  MEG Phantom experiment

#### H.3.1  Phantom Elektra

Dipoles in $m = 32$ various locations are emitting the same signal. Signal magnitude can be either very high, high or low, leading to 3 datasets: a very clean one, a clean one and a noisy one. These datasets are available as part of the Brainstorm application [55]. We preprocess the data using Maxwell filtering and low-pass filtering as done in the MNE tutorial `https://mne.tools/0.17/auto_tutorials/plot_brainstorm_phantom_elekta.html` and only consider data recorded by the magnetometers. We use the very clean dataset to recover the true signal by PCA with 1 component. Then we reduce the noisy dataset by applying view-specific PCA with $k = 20$ components and algorithms are applied on the reduced data. We select the component that is closer to the true one and compute the L2 norm between the predicted component and the true one after normalization. Then we attempt to recover the position of each dipole by performing dipole fitting on the mixing operator of each view (using only the column corresponding to the true component). The localization error is defined as the mean l2 distance between the true localization and the predicted localization where the mean is computed across dipoles. Each epoch corresponds to 301 samples and 20 epochs are available in total. We vary the number of epochs between 2 and 18 and display in Fig 7 the reconstruction error and the localization error in function of the number of epochs used. ShICA-ML outperforms other methods. ShICA-J gives satisfying results while being much faster.

#### H.3.2  Phantom Sinusoidal components

For completeness, we display the results obtained on another MEG dataset where the true component is a known sinusoidal and $m = 8$ different locations are considered for the dipoles. We vary the number of epochs between 2 and 16 and display in Fig 8 the reconstruction error and the localization

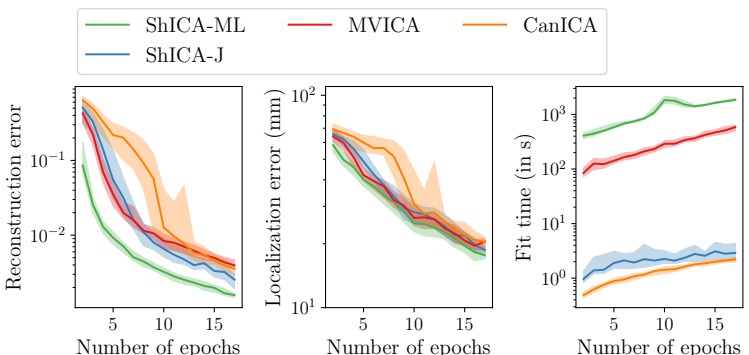

Figure 7: **MEG Phantom (Elektra)**: (left) L2 distance between the predicted and actual component (middle) Mean error (in mm) between predicted and actual dipoles localization (right) Fitting time (in seconds)

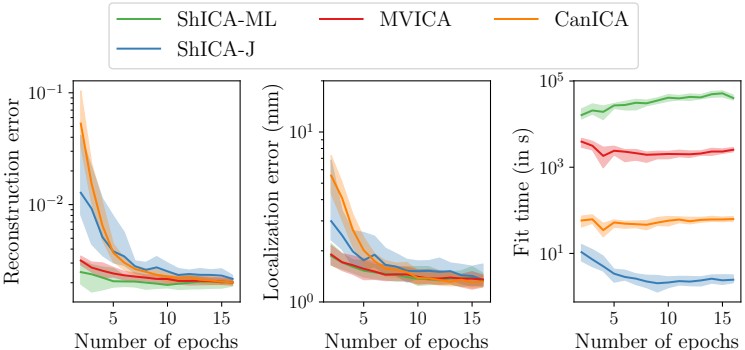

Figure 8: **MEG Phantom Sinusoidal components**: (left) L2 distance between the predicted and actual component (middle) Mean error (in mm) between predicted and actual dipoles localization (right) Fitting time (in seconds)

error as a function of the number of epochs used. ShICA-ML outperforms other methods. ShICA-J gives satisfying results while being much faster.

## H.4 CamCAN MEG components

We consider the CamCAN dataset used to produce Fig 4. We use $m = 496$ subjects and fit ShICA-ML with $p = 10$ components. We localize the components of each subject using sLoreta [44]. Then components are registered to a common brain and averaged. Thresholded maps are displayed below along with the time courses of each component. Components obtained with ShICA-ML highlight the ventral visual cortex and auditive cortex. The results suggest that the response of the auditive cortex is faster and lasts a shorter time than the response of the ventral visual cortex.

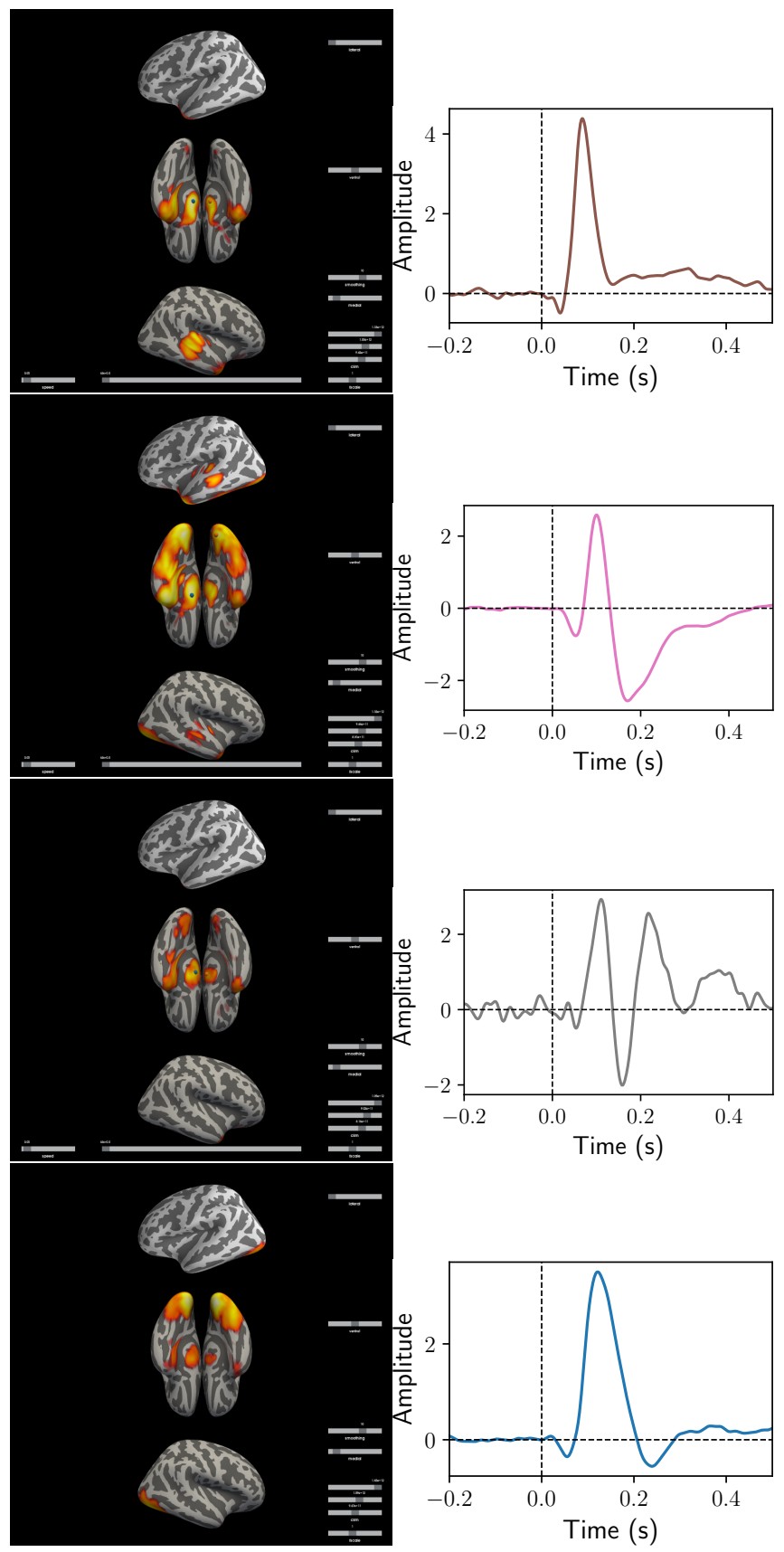

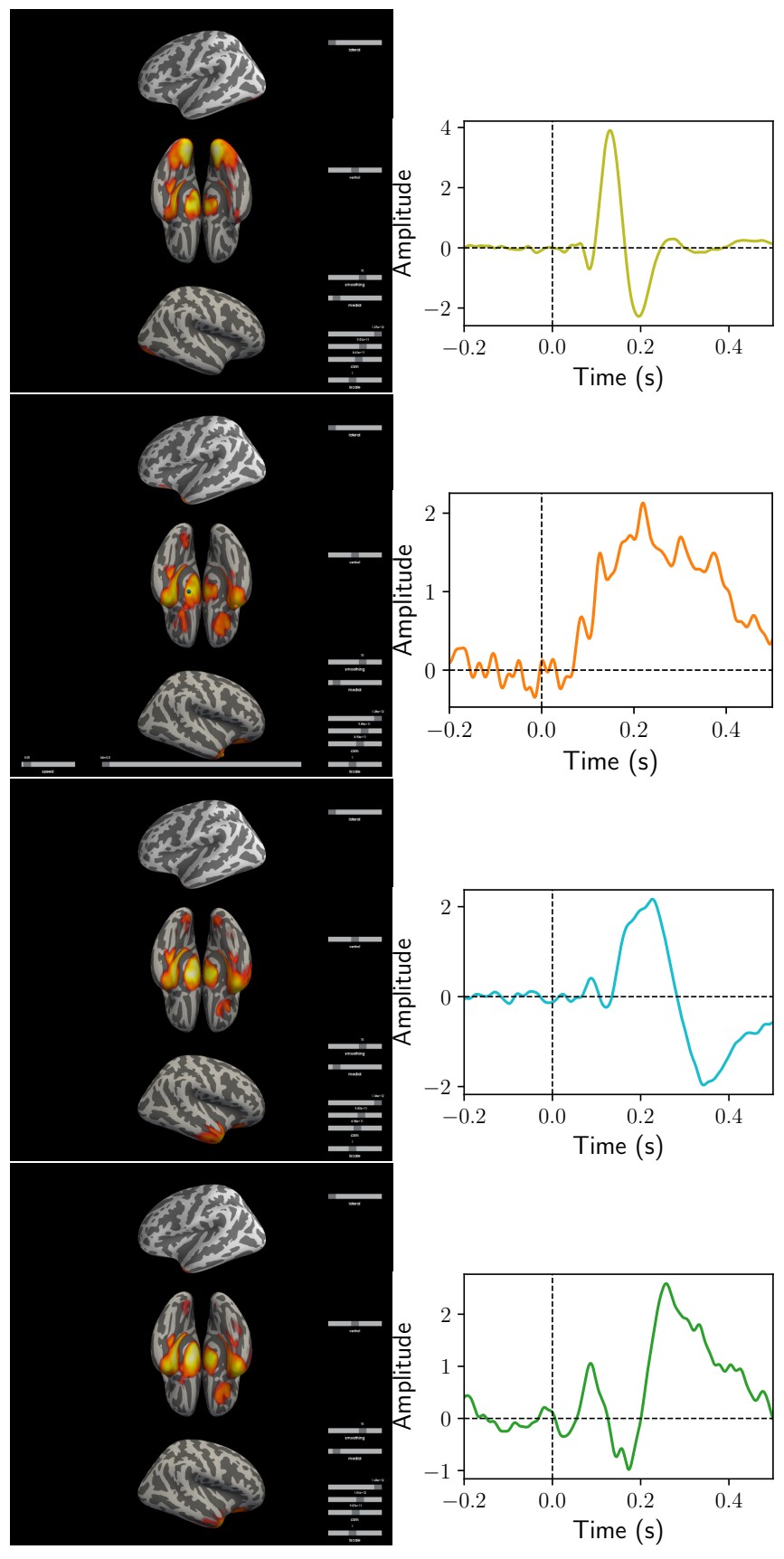

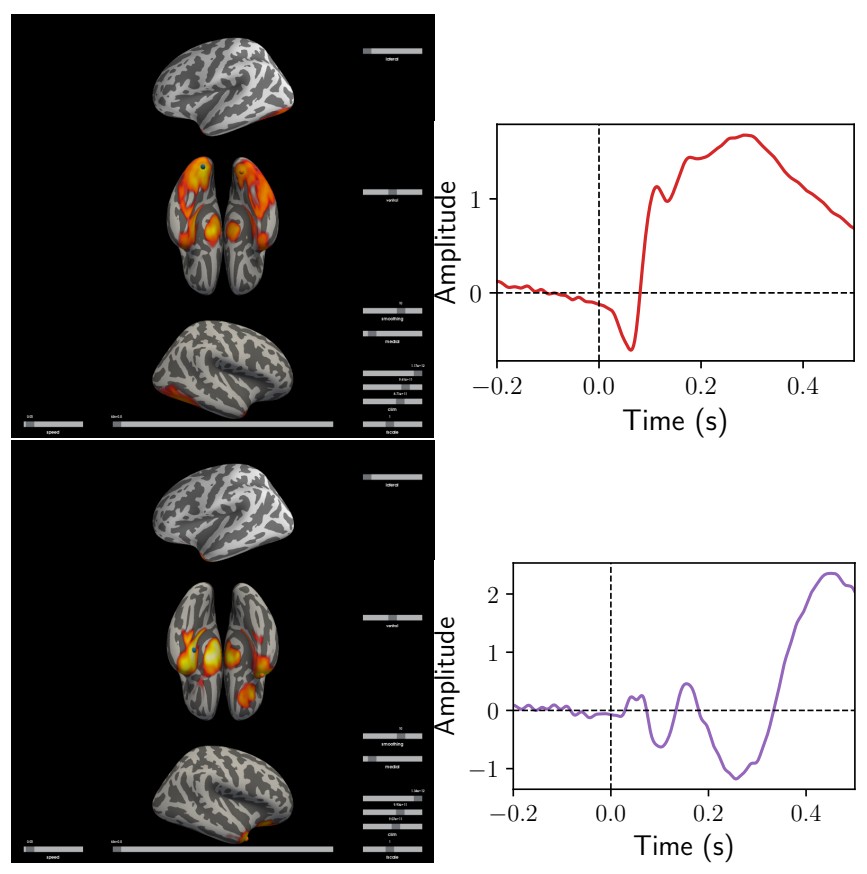