# OpenReview forum: "Shared Independent Component Analysis for Multi-Subject Neuroimaging"
_NeurIPS.cc/2021/Conference — NeurIPS 2021 Poster_

### Official Review · Reviewer_LARb · 2021-07-03

**Rating:** 7
**Confidence:** 3

**Summary:**

This paper presents a method, called shared ICA, for finding shared independent components across multi-view neuroimaging data. ShICA models each dataset as a linear transform of shared independent components with Gaussian noise. Necessary and sufficient conditions for the identifiability of ShICA are first discussed. Then two variants of the implementation are introduced, where ShICA-ML is more robust than ShICA-J in terms of deriving unmixing matrices under perturbation. Experiments on MEG and fMRI show that ShICA shows potential improvement in reproducibility, time segment matching, and data reconstruction.

**Limitations And Societal Impact:**

See point 3 of the main review.

**Main Review:**

The paper provides necessary and sufficient conditions for the identifiability of the independent components under certain assumptions, which seems to be an improvement over prior works. The maximum-likelihood solution based on EM looks reasonable. The method section is drafted in a rather clear fashion. I now list my concerns in the following:

1. Formulation: the authors name the generative process of Eq 1 as ShICA, but this is the same formulation as in [41]. The difference, as the authors mention, is the non-Gaussianity, noise model, and inference method. This limits the novelty and impact of the work. Moreover, the practice of putting both view-specific difference and measurement noise into a big blender $n_i$ worries me as these two sources of noise clearly entail different structures. Modeling measurement noise before mixing rather than after mixing  (outside the parenthesis) is unnatural to me.

2. Validation:
a. ICA-based multi-modal analysis is a widely explored area, which goes beyond the SRM papers cited herein. The review paper by https://www.ncbi.nlm.nih.gov/pmc/articles/PMC4917230/ lists 10+ practices for multi-modal neuroimaging analysis. jICA, pICA, and linked ICA are all conceivably good counterparts to compare with. Although the experimental setup highly overlaps with [41], the present baselines are a just subset of the baselines in [41].

b. I'm a bit confused why different baselines are used for different figures, e.g., MVICA has a runtime measure in Figure 3, but does not have an accuracy measure in Fig. 2.

c. More details should be given for the MEG and fMRI experiments, at least in the supplement (study cohort description, data spatial/temporal resolution/size, experimental task design in fMRI, etc).

d. I totally do not understand the last three figures in the supplement, no captions given.

3. Impact on neuroimaging: the present experiments are all from a signal-processing perspective, e.g., reconstruction and segment mathcing. It is unclear how this method can eventually be used in neuroimaging studies. Well-established methods like groupICA can efficiently analyze hundreds of subjects in modern neuroimaging studies, which is unlikely to be feasible for ShICA due to the EM inference.

**Time Spent Reviewing:**

2.5 hours

---

> ### Author Response · Authors · 2021-08-10
> **Thank you.**
>
> Dear reviewer,
>
> We thank you for taking time to review our article and for your insightful comments.
>
> 1. We believe the formulation of our model equation (1) makes the link between our approach and the approach of [41] easier to understand. ShICA is a strict generalization of [41]. As you noticed, in contrast to [41], the noise variance is learned and is distinct across subjects and across components. This makes a huge difference because ShICA can more efficiently use second order statistics to separate the sources.
> The novelty of this work is also given by: the identifiability results, the theorem 5 that shows why multiset CCA should be able to separate sources, the study in section 3.2 showing why this fails in practice, the joint diagonalization trick that allows to solve the problems of multiset CCA, the minimum mean squared error estimates of section 3.3 and section 4. as well as the fast quasi-newton EM of section 4.
> Regarding the noise on the sources: the noise in our model is not measurement noise. The term “noise” may be better understood as “deviation from the mean”. It makes a lot of sense to assume that subjects might deviate differently from the mean response. For instance, the response of some subjects might be of larger magnitude than others making them deviate a lot more from the mean. This can depend on the network considered which is why the noise variance corresponding to different components may not be the same. We used the term “noise” because in our view, the quantity of interest is the shared response.
> Nevertheless, we agree that it would have been better to model both the noise on the sensor and the noise on the sources. However, this would have broken our fast estimation and inference methods (as an example the method in [33] uses noise on the sensors and as described in l238-240 this leads to an intractable E-step), and greatly complicated the theory and the algorithms.
>
> 2.a We focused on a selection of methods that use non-Gaussianity and assume different mixing matrices but the same sources. jICA uses instead a shared mixing matrix and pICA (Beckmann, C. F., & Smith, S. M. (2004). Probabilistic independent component analysis for functional magnetic resonance imaging. IEEE transactions on medical imaging, 23(2), 137-152) is a single subject method - I believe you meant its extension to multiple subjects: TensorICA (Beckmann, C. F., & Smith, S. M. (2005). Tensorial extensions of independent component analysis for multisubject FMRI analysis. Neuroimage, 25(1), 294-311.). We could have compared our approach with linked ICA as well as with TensorICA. While we believe a comparison with the methods we missed is interesting, we believe the benchmarks we performed are extensive and convincing enough.
>
> 2.b If you look at the Figure 7 in appendix, you’ll see the results with MVICA. What we see there is that MVICA can also to some extent separate Gaussian sources even though the performances are not as good as ShICA-ML. After reading your comment, we think it would be better to put Figure 7 in the main text instead of Figure 2.
>
> 2.c In appendix F, some information about the datasets are available but others are missing indeed.
> Descriptions of the study cohort and experimental design are too long to be included in the rebuttal. We refer the reviewer to [12] for Sherlock and [23] for Forrest. The cohort description for Raiders and Gallant is available in [40]. The protocol used for Raiders is the same as the one used in (Haxby, Neuron 2011) and the protocol used for Gallant is the same as the one used in (Nishimoto, Current Biology 2011). We will include the corresponding references in appendix F.
> We have used the same full brain mask for all datasets. The mask contains 212445 voxels so this is the spatial dimensionality for all fMRI datasets. The number of subjects is given in appendix F. The TRs for Sherlock is 1.5s and 2s in other fMRI datasets. The Forrest dataset has a spatial resolution of 1mm while other fMRI datasets have a spatial resolution of 3mm.
> The Forrest dataset uses 7 runs of respectively 451, 441, 438, 488, 462, 439 and 542 timeframes and 19 subjects (we said 12 in appendix F this is a mistake).
> The Raiders dataset uses 9 runs of respectively 374, 297, 314, 379, 347, 346, 350, 353 and 211 timeframes.
> The Gallant dataset uses 17 runs of 325 timeframes.
> About the datasets used in MEG, we only consider the magnetometers so the dimensionality is 102 in all datasets. The protocol used in the CamCAN dataset is described in [45]. The number of subjects used is described in the text. After splitting the data as described in the paragraph l358 we get 6 chunks each containing 500 samples of signal (where subjects are exposed to the stimuli) and 200 samples of rest (where no stimuli is displayed). The temporal resolution is 1ms.
> The Phantom datasets used in the appendix also come with a temporal resolution of 1ms. Each epoch represents 301 samples and the number of epochs used are described in appendix G.3.
> Above information will be added to appendix F.
>
> 2.d The three last figures represent the shared sources and their localization on the brain obtained while subjects were exposed to an audio visual stimuli (see section G.4 for a more complete description).
> We believe that the lack of clarity comes from Figure 10 being between the text of section G.4 and the last figures. We will re-organize this section.
>
> 3. We believe ShICA is a scalable method.  In appendix G.4 we fit ShICA-ML on the CamCAN datasets using  496 subjects. The closed form E-step makes the EM rather fast. But ShICA-ML is slower than GroupICA for sure.
>  ShICA-J on the other hand is as fast as GroupICA (if not faster since joint diag can be faster than ICA - this depends on the number of samples).

---

### Official Review · Reviewer_qGdn · 2021-07-13

**Rating:** 7
**Confidence:** 5

**Summary:**

This paper introduces a new generative model and inference algorithms for shared Independent Component Analysis (ShICA). The model incorporates an explicit noise term on the components, which allows them to be identifiable up to a sign and permutation. The authors provide conditions to guarantee identifiability and introduce two solution algorithms. ShICA-J assumes Gaussian components and relies on multiset CCA coupled with a joint diagonalization. ShICA-ML assumes a specific non-Gaussianity and relies on the EM algorithm for component estimation. The authors demonstrate the robustness of their algorithms on simulated data and do an exploratory evaluation on MEG and fMRI.

**Limitations And Societal Impact:**

The authors acknowledge that their framework cannot perform dimensionality reduction and relies on a simple non-Gaussianity. However, they do not discuss other potential limitations (both theoretical and empirical) described above in the “Main Review”.

While this work has natural application to functional neuroimaging, the performance gains are not particularly impressive, particularly for fMRI. The societal impact sentence is generic and overly broad. Given that decades of research involving PCA, ICA, dictionary learning, deep learning, etc. has made a minimal impact on “reducing the human and societal burden of brain disease”, it is unlikely that, yet another variant of ICA will accomplish this feat.

**Main Review:**

Overall, this is a solid paper that presents an interesting new twist on ICA. The authors complement the model with theoretical guarantees for identifiability, assuming specific noise characteristics. With that said, the assumptions seem overly restrictive, and the performance gain is marginal on real-world data. Specific comments on the paper are as follows:

Strengths

+ The framework imposes an explicit noise model on the components, which allows for cross-subject fusion and confidence estimation. The noise model also provides identifiability of the components.
+ The authors introduce two estimation algorithms that tradeoff efficiency and generalization.
+ Theoretical guarantees are provided for each optimization procedure.
+ The simulated experiments are quite extensive and clearly demonstrate the performance regimes for which each method (introduced and baselines) is well suited.
+ The MEG also demonstrates the robustness of the SHICA-ML method over the others.

Weaknesses

- Assumption 1 (noise diversity) seems fairly restrictive, and it is unclear from a modeling standpoint, why one would assume this for real-world data. Taking fMRI as an example, the components would roughly correspond to different functional networks in the brain. Why would we assume a priori that one network is “noisier” than another? Even if this were the case empirically, to strictly require it seems presumptive.
- As far as I can tell, there is no requirement that the mixing matrices A_i be similar across the views. This seems problematic, as I would assume the different “views” in an MEG or fMRI experiment correspond to different subjects. Thus, one would want to identify similar activations patterns across the group.
- While Section 3 (ShICA-J) is interesting from an algorithmic standpoint, it is unclear how useful this algorithm is in practice. Specifically, ShICA-J assumes only Gaussian components, which is contrary to one of the main advantages of ICA in general (non-Gaussianity). At a high level, ShICA-J is closer to a PCA type analysis than the standard ICA. I would have liked to see a comparison with common PCA.
- It is unclear why the authors claim that most non-Gaussian components in real data are super-Gaussian, or why they assumed the particular density in Section 4. At the very least, it seems that the mixture weights and variances of the non-Gaussian density should be estimated via EM.
- The BOLD reconstruction results are very similar across methods. From a practical standpoint, there is little difference between the reported R^2 values. Plus, the confidence intervals are highly overlapping.

**Time Spent Reviewing:**

3 hours

---

> ### Author Response · Authors · 2021-08-10
> **Thank you.**
>
> Dear reviewer,
>
> We thank you for taking time to review our article and for your insightful comments.
>
> 1. (assumption 1 is too restrictive) In that context, the term “noise” may be better understood as “deviation from the mean”, in particular due to individual differences in the brains. It makes a lot of sense to assume that subjects might deviate differently from the mean response. For instance, the response of some subjects might be of larger magnitude than others making them deviate a lot more from the mean. This can depend on the network considered, which is why the noise variance corresponding to different components may not be the same. We used the term “noise” because in our view, the quantity of interest is the shared response.
> Let us stress that in the context of this paper the shared sources are temporal and not spatial. While we believe it would also make sense to perform spatial ICA using ShICA, the underlying hypothesis about the signal are quite different. As an example, we believe it would make sense to analyze resting state data with ShICA if one is interested in finding shared spatial topographies across subjects, but this is left for a future study.
>
> 2. (no requirements that mixing matrices A_i be similar) We believe this really depends on what the goal of your study is. In our experiments Figure 4,5 or 8, healthy subjects are exposed to the same naturalistic stimuli. We want to recover a stereotypical response and we are also interested in the way  the spatial organization of their networks differs. Indeed components in each subject are synchronized so that column j of A_i gives insight about the spatial organization in subject i corresponding to the j-th time course in S.
>
> 3. (benchmark PCA) We have actually benchmarked Group PCA (where you concatenate all views feature-wise, perform PCA, and recover unmixing matrices) on the experiments in Figure 2 (although this was not included in the paper). PCA can separate neither Gaussian nor non-Gaussian sources even with noise diversity. This is expected: PCA selects the sources of maximal power, while ShICA-J actually finds independent sources. Let us also highlight that in general the stacked mixing matrices have no reason to form an orthogonal matrix. The fact that ShICA-J or Multiset CCA can perform some kind of source separation is striking and we believe this is one interesting contribution of this paper to show clearly why (Theorem 5). We should also note that the ability to separate even Gaussian sources is a benefit of the algorithm compared to non-Gaussian methods: indeed, if two independent sources are non-Gaussian and also have noise diversity, then ShICA-J will be able to separate them: ShICA-J can separate sources with any density, as long as the noise diversity hypothesis is satisfied.
>
> 4. (super Gaussian components in real data) Indeed a citation is missing here. We rely on the work of (Delorme, A., Palmer, J., Onton, J., Oostenveld, R., & Makeig, S. (2012). Independent EEG sources are dipolar. PloS one, 7(2), e30135.) and (Calhoun, V. D., & Adali, T. (2006). Unmixing fMRI with independent component analysis. IEEE Engineering in Medicine and Biology Magazine, 25(2), 79-90.) to say that most brain sources are super-Gaussian. We will add this reference.
> We chose a super-gaussian density for this reason. The Gaussian mixture is also particularly easy to work with and optimize. However we believe that optimizing the mixture weights and variances would be possible while still keeping an exact E-step and a fast M-step. Our sentence l305-306 “modeling the non-Gaussianity in more details in ShICA-ML should improve performance” should be understood in this way.
> Yet, in practice, ShICA-ML can separate superGaussian and subGaussian sources (see results of Figure 6 in appendix - although this may be only possible with noise diversity, we did not try without). A great thing with ICA in general is that it is not necessary to identify the exact density for the algorithm to work. For example, in classical ICA, we can separate sources with a super Gaussian density using a model with a different super Gaussian density (Cardoso, J. F. (1998). Blind signal separation: statistical principles. Proceedings of the IEEE, 86(10), 2009-2025.).
> Note that in our synthetic experiments, we never used the density of the sources in the model but always a different one.
>
> 5. (Improvement in Figure 5 is low) Indeed, but the improvement is consistent over datasets and the number of components. While we agree that the improvement is modest in this experiment, we believe it still shows that our method is slightly better for this task. The results of the additional experiments Figure 8, 9 and 10 in appendix also show that ShICA-ML yields better results than competitors.
>
> 6. (“reducing the human and societal burden of brain disease”) This last sentence is indeed a bit too optimistic. We will replace it with a more thorough discussion about the limitations of the current work.

---

> > ### Comment · Reviewer_qGdn · 2021-08-29
> > **Rebuttal**
> >
> > Thank you for the thoughtful response to my comments.

---

### Official Review · Reviewer_9MQr · 2021-07-14

**Rating:** 7
**Confidence:** 4

**Summary:**

The paper introduces Shared Independent Component Analysis (ShICA), a multi-view ICA method which assumes each view as a linear transformation of a single set of shared independent components with additive gaussian noise. It can be viewed as a generalization of a slew of "component analysis" methods such as GroupICA, Multiset CCA, Independent Vector Analysis, and SRM. Unlike GroupICA methods, ShICA allows for Gaussian (as well as non-Gaussian) components, different noise covariance and extraction of shared components. While under certain conditions Multiset CCA can recover the mixing matrices in ShICA formulation, the paper proves conditions where that is not the case, and demonstrates the lack of robustness of Multiset CCA in certain situations. ShICA is also closely related to IVA, but allows for extraction of shared components. Similarly, it can be viewed as a relaxation of (biologically infeasible) orthogonality constraint on SRM.

The paper provides assumptions and theorems laying out the theoretical framework for ShICA and the inference algorithms used. It also theoretically demonstrates the connection between ShICA and Multiset CCA, and uses joint diagonalization of multiset CCA solution to recover unmixing matrices in situations where multiset CCA can't. For situations with gaussian and non-gaussian components the paper introduces ShICA-ML a maximum likelihood general estimator for ShICA. Experiments included simulations as well as fMRI and MEG datasets.

**Limitations And Societal Impact:**

are addressed appropriately

**Main Review:**

Overall the paper is well-written and does a good job backing up claims with theory as well experiments. While the problem and formulation is related to previous work, the paper clearly lays out where it differs and improves upon the existing work. I think this is a significant contribution to the independent component analysis literature. I'd however like to see responses to the following concerns. please note that these are ordered in the order they appear in the paper, I will mark the major concerns as such, but would appreciate if the authors can respond to all/ as many of these as possible.

- (Major) Title: the paper is titled "Shared Independent Component Analysis for Multi-Subject Neuroimaging". While I'm all for interdisciplinary work that is theoretical but also attack specific applications. I found this title to be somewhat misleading. While it's true that the paper uses fMRI data and MEG data in the experiments there is really no significant discussion of why this method is specifically geared towards neuroimaging. For example, the words neuroimaging and neuroscience appear literally once in the main paper.  One possible solution is to expand upon the neuroscientific implications a little bit more in the first paragraph (as opposed to just the one line that's there), or add more discussion in the conclusion section.

-line 39 "each dataset is modelled as..." somewhat confusing use of the word dataset here. is x_1,...,x_m being referred to as one dataset, or x_1 is one dataset?

- (Major) lines 47-54. The line starting "we then introduce an algorithm called ShICA-J..." and the line starting "We next point out the practical problem..." make it sound like these are two different contributions. From reading the paper (and from Algorithm 1) what I gathered is that ShICA-J is the joint-diagonalization of the W_i's obtained from Multiset CCA and is the solution to the sampling noise leading to large rotations in Multiset CCA solution. The way these lines are written right now makes it sound like ShICA-J is separate from joint diagonalization of the result of multiset CCA.

- (Minor) line 91. The theorem/assumptions/propositions are all being counted in a single series. I think it will improve readability if you can separate out the numbering for theorems from numbering for assumptions etc... The way it is right now I went looking for Theorem 1 (which of course doesn't exist)

- line 113. "Generalized CCA consists in" --> "Generalized CCA consists of"  .

- line 224. Citation required for "most non-Gaussian components in real data are super-Gaussian"

following concerns regarding experiments, while not very major, will definitely go a long way in convincing me to possibly increase my score for this paper:

- Figure 1: I had somewhat of a hard time with the figure. It might just need a clearer legend. What is the black dashed line at 10^-1? just like there are 4 solid colored lines, shouldn't there be 4 dashed colored lines corresponding to the different eigen gaps with joint diagonalization solution?

- line 299. "Deep variants of SRM... are much more computationally demanding". Would be nice to see some experiments showing the truth of this statement. Or if this is established based on experiments in some other publication please provide citation.

- Figure 2: why are results for MVICA not included in this figure?

- line 373: Please include the original dimensionality of the data as well and why was p=10 components choosen?
- line 387: what was the number of factors used in SRM preprocessing and why?, what was the original dimensionality?
- (Major) Figure 5: This connects to my comment about the title. These results don't seem all that much of an improvement. In terms of R^2 score except for "gallant" dataset all methods are pretty much behaving in a very similar manner while CanICA seems to have a lower fit time in most cases. Now if this were just one of the many applications shown in experiments that would have been okay, but since the title is gearing me to expect good performance "specifically" on neuroimaging data, these results seem somewhat underwhelming. I'm happy to be convinced otherwise, else the title/intro/conclusion changes become all the more important.

**Time Spent Reviewing:**

8

---

> ### Author Response · Authors · 2021-08-10
> **Thank you.**
>
> Dear reviewer,
> We thank you for taking time to review our article and for your insightful comments.
> 1. (misleading title) Thanks for this remark. This method was developed with neuroimaging applications in mind but it is presented in a theoretical way so that people outside neuroimaging can more easily use our work. But we may have gone a bit too far in this direction.
> Initially, ShICA was developed to recover shared components from the data of subjects exposed to naturalistic stimuli, where the functional organization of each subject is different and the magnitude of the response might also differ. ShICA allows to tackle both of these issues. We believe this has strong implications in neuroscience as ShICA can be used to reduce inter-subject variability, obtain cleaner and more stable common sources and efficiently transfer information across datasets that share the same subjects.
> We will follow your suggestion and add more insights in the introduction and the discussion section.
> 2.  (comment on l39) x_1 is one dataset here. As a side note the term “datasets”, “views” and “subjects” mean the same thing here. We tried to use “view” everywhere but it appears that we missed this one.
> 3. (Description of ShICA-J does not match Algorithm 1) You understood correctly. We will rewrite this part in order to make it clear that ShICA-J is what is described in Algorithm 1.
> 4. (numbering of theorems and assumptions) Agreed, we will change the numbering as requested
> 5. (missing citation for super Gaussian sources)
> Indeed. We will add (Delorme, A., Palmer, J., Onton, J., Oostenveld, R., & Makeig, S. (2012). Independent EEG sources are dipolar. PloS one, 7(2), e30135.) and (Calhoun, V. D., & Adali, T. (2006). Unmixing fMRI with independent component analysis. IEEE Engineering in Medicine and Biology Magazine, 25(2), 79-90.).
> 6. (unclear Figure 1) Indeed this may not be so clear. The black dashed line around 10^-1 is the performance at chance level (we can see this as the worst possible performance). We will make that clear in the text.
> What is so great with this experiment is that the 4 dashed colored lines actually coincide. So even if the eigengap is very small, and the perturbation much larger than the eigengap, the joint diagonalization allows to recover the correct solution. We will clarify this in the caption of the figure.
> 7. (justify that deep variant of SRM are slow) We did not benchmark the deep variants of SRM. The authors of the deep hyperalignment (Yousefnezhad, NeurIPS 2017) benchmark some deep variants on small brain regions and claim that their method is only slightly slower than SRM. The “much more” may therefore be too strong although we would find it quite surprising that a network using 3 hidden layers with a number of units of the order of the number of voxels can ever be faster than SRM on full brain data. We will rephrase to highlight instead that our method is more easily interpretable, easier to train and does not have as many hyper-parameters to tune.
> 8. (MVICA missing in Figure 2) If you look at the Figure 7 in appendix, you’ll see the results with MVICA. What we see there is that MVICA can also to some extent separate Gaussian sources even though the performances are not as good as ShICA-ML. After reading your comment, we think it would be better to put Figure 7 in the main text instead of Figure 2.
> 9. (dimensionality of the data and number of components) The original dimensionality of the data is 102 (102 magnetometers). We think it would also make sense to perform this study with 20 components for example. We chose 10 components in order to have a faster computation time. We will add this information in the text.
> 10. (number of components in SRM) We used SRM only in fMRI experiments (otherwise the pre-processing is performed with PCA). We used full brain data so the original dimensionality is given by the number of voxels in the mask: 212445. The number of components is indicated in Figure 5 (ShICA - like most ICA methods - can only work on reduced data).
> 11. (results of Figure 5 are underwhelming) Indeed the improvements are modest but we would like to emphasize that they are systematic. We believe this shows that the method works better than competitors even if the improvement is not so impressive.
> Note that this is not the only experiment we have made. In addition to the results of Figure 4, you can also look at Figure 8, 9, 10 for further experiments on real data.  We believe that the improvement on MEG Phantom data (Figure 8 and 9) is more impressive.

---

> > ### Comment · Reviewer_9MQr · 2021-08-23
> > **Satisfied with the response.**
> >
> > I am satisfied with the response provided by the authors and will be upgrading my evaluation to 7: Accept.
> > Dear authors, please make sure the promised changes are made.

---

### Official Review · Reviewer_GxXa · 2021-07-16

**Rating:** 7
**Confidence:** 4

**Summary:**

The paper considers a multi view (or multi subject) ICA model and provides a few methodological extensions and results. The Gaussian source assumption, EM optimization, and connections to mCCA + joint diagonalization for efficient initialization are the main contributions relative to the previously published model [41]. This work also supports different noise variances in different sources and different views, and presents a stronger (global) identifiability proof. The work is largely inspired by the temporal group ICA and is presented in that paradigm: shared timecourses, assumed shared task stimuli across subjects etc.


**Limitations And Societal Impact:**

no limitations

**Main Review:**

## The main concerns:

1.  Overall, the paper heavily builds on reference [41] and the previous work is not discussed enough thus misrepresenting the context of the current work.
2.  The papers expects the underlying system to consist of shared sources (here time courses) well modeled by ShICA. This feature is used to justify invalidity of IVA for ALL of fMRI analysis. Even under identical stimuli, a shared (identical) response over subjects is all but a fictional (though useful) construct, which cannot be the basis for establishing the utility of a method. Consider that the entire field of Spatial ICA for fMRI is fervently moving AWAY from shared sources in order to better model subject variability. Furthermore, it is not inconceivable that certain brain networks are highly variable from subject to subject to the point they may not even correlate at all and, even more, they can be dynamic (changing) over space AND time. The more you think about it, the less likely it is that a shared source model is true, even in task-based settings, when we consider mind wondering, task engagement issues, and various mental illnesses. It would be more reasonable to not criticize IVA for "lack" of a shared construct, and instead assess it on the basis of captured source variability at the subject level, i.e., compare s<sub>ij</sub> from IVA to (s<sub>j</sub> + n<sub>ij</sub>) of ShICA.
3.  Related to the above comparisons are to the outdated IVA-L (no correlation estimation) and IVA-G (no HOS estimation). It is not unreasonable to apply the same joint diagonalization trick from ShICA-J to “fix” IVA-G, for example. And IVA has evolved since 2012.
    1.  <https://ieeexplore.ieee.org/document/6638257?arnumber=6638257>
    2.  see also IVA-L-SOS and IVA-GGD in <http://mlsp.umbc.edu/jointBSS_introduction.html>
4.  Note a contradiction in the claims. First the paper claims that "… These methods are weaker than ShICA as they assume non-Gaussianity, the same noise covariance across views and lack a principled method for shared response inference" Then it adds: "… ShICA-ML uses a simple model of a super-Gaussian distribution, while modelling the non-gaussianities in more detail in ShICA-ML should improve the performance" But ShICA-ML is the best model, even though it uses non-gaussianity: "We further introduce ShICA-ML, a maximum likelihood estimator of ShICA that models non-Gaussian components using a Gaussian mixture model. While ShICA-ML yields more accurate components, ShICA-J is significantly faster and offers a great initialization to ShICA-ML".
5.  Experimental results are incomplete and not just for lacking comparison with the latest IVA algorithms. The MultiView ICA [41] is not compared against. Some of the figures list MVICA, which be it, but the acronym is neither defined not properly discussed. Additionally if one was to compare Figure 5 in the current submission with Figure 2 from [41] - MultiView ICA looks extremely competitive if not better. The MVICA lines for the same datasets in Figure 5 are drawn lower than the model, if they are indeed the same models, was capable of in [41]. Additionally, the accuracy plot is missing in Figure 5 and that was the most discriminative plot of Figure 2 in 41.

**Time Spent Reviewing:**

5

---

> ### Author Response · Authors · 2021-08-10
> **Thank you.**
>
> Dear reviewer,
> We thank you for taking time to review our article and for your insightful comments.
> 1. While the formulation of our model equation (1) makes the link between our approach and the approach of [41] easier to understand, ShICA is a strict generalization of [41]. As you noticed, in contrast to [41], the noise variance is learned and is distinct across subjects and across components. This makes a huge difference because ShICA can more efficiently use second order statistics to separate the sources.
> The novelty of this work is also given by: the identifiability results, the theorem 5 that shows why and under which conditions multiset CCA should be able to separate sources, the study in section 3.2 showing why this fails in practice, the joint diagonalization trick that allows to solve the problems of multiset CCA, the minimum mean squared error estimates of section 3.3 and section 4. as well as the fast quasi-newton EM of section 4. While some of the experiments are the same as in [41] the experiments in Figure 2, 3, 4, 6, 7, 9 are new. For this reason we believe that the novelty with respect to [41] is clearly demonstrated.
> In order to better represent the context of the current work, we will include in the related work section a discussion about the more recent IVA techniques (IVA-L-SOS, IVA-GGD and IVA with Kotz distribution). We will state that these methods should be able to use SOS and HOS to separate sources like ShICA-ML.
>
> 2. The lines  (l35-36) in the introduction are indeed a bit unfair to IVA. We do not believe that IVA is invalid for all of fMRI analysis. However we believe that the ability of ShICA to extract a shared response is definitely a feature: it highlights a stereotypical brain response to a stimuli. Further on that point,this is also what methods such as hyperalignment aim at: reducing inter-subject variability means finding what is shared. This being said, we definitely agree that in some contexts, we might want to study what varies between subjects: this is in fact an important feature of our method, where the noise term also models the individual differences of the responses, in addition to measurement noise. We will rewrite this part of the intro to reflect that IVA and ShICA are both useful methods but in different contexts.
>
> 3. Thanks for these references. Indeed in the related work we mostly criticize IVA based on IVA-L and IVA-G because these methods are the most popular (probably because they are the oldest).  We agree that IVA-L-SOS, IVA-GGD and IVA with Kotz distribution should be able to use SOS and HOS like ShICA-ML do. We will mention these 3 methods in the related work section. As a side note, the additional parameter Sigma in IVA with Kolz distribution is not updated according to the maximum likelihood estimator. We therefore believe the optimization procedure used in ShICA-ML is slightly cleaner.
>
> 4. Our phrasing might be a bit unclear here. The important point is that in contrast to MultiView ICA [41] or the approach of Guo [22], we explicitly model the noise covariance. Therefore we do not only use non-Gaussianity but also noise diversity. We will clarify this in the text. We will rephrase to say that these methods need non-Gaussianity for identifiability while in ShICA, even Gaussian sources can be identified.
>
> 5.The fact that the acronym MVICA is undefined is an error from our part: we will make it clear that it is MultiViewICA. Figure 5 and Figure 2 from [41] should be the same: we actually used the Github implementation they published with their paper.
> In order to make things clearer: we propose to add, in the supplementary material, a table that gives the median R2 score and the confidence interval reported in Figure 5 for MVICA and ShICA using 20 components:
>
> ```
> dataset  | method   | R2 score | Confidence interval
> forrest  | ShICA-ML | 0.200    | [0.187, 0.213]
>          | ShICA-J  | 0.171    | [0.157, 0.185]
>          | MVICA    | 0.191    | [0.177, 0.204]
> gallant  | ShICA-ML | 0.121    | [0.107, 0.135]
>          | ShICA-J  | 0.110    | [0.095, 0.125]
>          | MVICA    | 0.114    | [0.099, 0.128]
> raiders  | ShICA-ML | 0.158    | [0.142, 0.174]
>          | ShICA-J  | 0.146    | [0.129, 0.162]
>          | MVICA    | 0.144    | [0.124, 0.164]
> sherlock | ShICA-ML | 0.174    | [0.157, 0.191]
>          | ShICA-J  | 0.165    | [0.146, 0.183]
>          | MVICA    | 0.161    | [0.142, 0.180]
>
> ```
>
>
> The accuracy plot is available in Figure 8 in supplementary material: MVICA and ShICA have similar performance on this task.
> It is not so straightforward to perform our experiments with IVA because IVA lacks the concept of shared response. For example, in the reconstruction experiment, you reconstruct the data of a left-out subject from the shared response and the mixing matrix of the left-out subject. Of course, estimates such as the mean of the subject specific responses could be used but it is heuristic. To us, current implementations of IVA solve a distinct problem.

---

> > ### Comment · Reviewer_GxXa · 2021-08-26
> > **good paper that can benefit from some tweaks**
> >
> > I have a few points remaining after the author's response that would be good to take care of in the manuscript.
> >
> > 1. It would help the readers to clarify early on the distinct target application that motivates the source noise model (i.e., clearly define the “contexts” in which a shared response model (ShICA) or linkage model (IVA) would be preferred). Specifically, because this is a (naturalistic) task fMRI temporal ICA, there is a strong expectation that the temporal sources will indeed be shared (bc same stimuli), whereas that is not necessarily be the case for spatial fMRI. This clarification is also important bc if there is no strong driver of shared spatial landmarks, like in spatial ICA, a shared response model is likely inappropriate bc, at that point, the source noise becomes the component of interest, and it is only Gaussian by definition.
> >
> > 2. Please include pre-processing steps in the code if you already did not.
> >
> > 3. mCCA: IVA-G is a generalization of mCCA (using GENVAR cost [1]) that allows non-orthogonal mixing. From this perspective, it seems contradicting to emphasize mCCA in this work and dismiss IVA-G as not useful in this (shared temporal ICA) context. It may be better to be less contrastive and provide a more unifying view of these two methods.
> > Does the separation proof for mCCA extend to IVA-G as well?
> > [1] http://dx.doi.org/10.1007/978-3-642-15995-4_44
> >
> > 4. The fast quasi-newton EM of section 4 has been generally described in the textbook “Handbook of Blind Source Separation”, Ch. 4 from Jean-Francois Cardoso. This reference is missing
> >
> > I am bumping my score to simplify the decision.

---

### Decision · Program_Chairs · 2021-09-28

**Decision:**

Accept (Poster)

**Comment:**

This paper introduces Shared Independent Component Analysis (ShICA), a method for shared response modeling in which shared components are independent. While building on previous work, this paper has clear novelty in model specification and related theory, using  multiple estimation algorithm with different noise specifications, and convincing experimental results.

The reviewers have proposed multiple changes that would improve the paper and that the authors have consented to do. Namely, a change in the framing, including a better discussion of the relevance to neuroscience and the relation to prior work, multiple clarifications, including in respect to model formulation and assumptions. I recommend acceptance for the paper and ask the authors to make these changes that will further improve its quality.

**Consistency Experiment:**

NeurIPS has a long history of experimentation. In 2014, NeurIPS ran an experiment in which 10% of submissions were reviewed by two independent committees to quantify the randomness in the review process. This year, we repeated a variant of this experiment to see how the quality of the review process has changed over time.  This paper was part of the experiment and was therefore assigned to two committees (consisting of reviewers, an Area Chair, and a Senior Area Chair) that reached independent decisions.  If both committees made the same recommendation, this recommendation was followed. If a single committee recommended acceptance, the paper was accepted (with the exception of a few cases in which the other committee identified what we considered a fatal flaw, e.g., an error in a key result).

Both committees reached the same decision: **Accept (Poster)**

The other committee assigned to the paper recommended **Accept (Poster)**.  You can find the other set of reviews, along with any follow up discussion with the authors here:
https://openreview.net/forum?id=yRTebElmilN